# Adversarial Vulnerability of Neural Networks Increases with Input Dimension

## Abstract

Over the past four years, neural networks have been proven vulnerable to adversarial images: targeted but imperceptible image perturbations lead to drastically different predictions. We show that adversarial vulnerability increases with the gradients of the training objective when viewed as a function of the inputs. For most current network architectures, we prove that the $\ell_1$-norm of these gradients grows as the square root of the input size. These nets therefore become increasingly vulnerable with growing image size. Our proofs rely on the network's weight distribution at initialization, but extensive experiments confirm that our conclusions still hold after usual training.

## 1 Introduction

Following the work of Goodfellow et al. (2015), Convolutional Neural Networks (CNNs) have been found vulnerable to adversarial examples: an adversary can drive the performance of state-of-the art CNNs down to chance level with imperceptible changes of the inputs. A number of studies have tried to address this issue, but only few have stressed that, because adversarial examples are essentially small input changes that create large output variations, they are inherently caused by large gradients of the neural network with respect to its inputs. Of course, this view, which we will focus on here, assumes that the network and loss are differentiable. It has the advantage to yield a large body of specific mathematical tools, but might not be easily extendable to masked gradients, non-smooth models or the 0-1-loss. Nevertheless, our conclusions might even hold for non-smooth models, given that the latter can often be viewed as smooth at a coarser level.

**Contributions.** More specifically, we provide theoretical and empirical arguments supporting the existence of a monotonic relationship between the gradient norm of the training objective (of a differentiable classifier) and its adversarial vulnerability. Evaluating this norm based on the weight statistics at initialization, we show that CNNs and most feed-forward networks, *by design*, exhibit increasingly large gradients with input dimension $d$, almost independently of their architecture. That leaves them increasingly vulnerable to adversarial noise. We corroborate our theoretical results by extensive experiments. Although some of those experiments involve adversarial regularization schemes, our goal is not to advocate a new adversarial defense (these schemes are already known), but to show how their effect can be explained by our first order analysis. We do not claim to explain all aspects of adversarial vulnerability, but we claim that our first order argument suffices to explain a significant part of the empirical findings on adversarial vulnerability. This calls for researching the design of neural network architectures with inherently smaller gradients and provides useful guidelines to practitioners and network designers.

## 2 From Adversarial Examples to Large Gradients

Suppose that a given classifier $\varphi$ classifies an image $\boldsymbol{x}$ as being in category $\varphi(\boldsymbol{x})$. An adversarial image is a small modification of $\boldsymbol{x}$, barely noticeable to the human eye, that suffices to fool the classifier into predicting a class different from $\varphi(\boldsymbol{x})$. It is a *small* perturbation of the inputs, that creates a *large* variation of outputs. Adversarial examples thus seem inherently related to large gradients of the network. A connection, that we will now clarify. Note that visible adversarial examples sometimes appear in the literature, but we deliberately focus on imperceptible ones.

**Adversarial vulnerability and adversarial damage.** In practice, an adversarial image is constructed by adding a perturbation $\boldsymbol{\delta}$ to the original image $\boldsymbol{x}$ such that $\|\boldsymbol{\delta}\| \leq \epsilon$ for some (small) number $\epsilon$ and a given norm $\|\cdot\|$ over the input space. We call the perturbed input $\boldsymbol{x} + \boldsymbol{\delta}$ an $\epsilon$-sized $\|\cdot\|$-attack and say that the attack was successful when $\varphi(\boldsymbol{x} + \boldsymbol{\delta}) \neq \varphi(\boldsymbol{x})$. This motivates

**Definition 1.** Given a distribution $P$ over the input-space, we call *adversarial vulnerability* of a classifier $\varphi$ to an $\epsilon$-sized $\|\cdot\|$-attack the probability that there exists a perturbation $\boldsymbol{\delta}$ of $\boldsymbol{x}$ such that

$$\|\boldsymbol{\delta}\| \leq \epsilon \quad \text{and} \quad \varphi(\boldsymbol{x}) \neq \varphi(\boldsymbol{x} + \boldsymbol{\delta}) . \tag{1}$$

We call the average increase-after-attack $\mathbb{E}_{\boldsymbol{x} \sim P}[\Delta \mathcal{L}]$ of a loss $\mathcal{L}$ the ($\mathcal{L}$-) *adversarial damage* (of the classifier $\varphi$ to an $\epsilon$-sized $\|\cdot\|$-attack).

When $\mathcal{L}$ is the 0-1-loss $\mathcal{L}_{0/1}$, adversarial damage is the accuracy-drop after attack. The 0-1-loss damage is always smaller than adversarial vulnerability, because vulnerability counts all class-changes of $\varphi(\boldsymbol{x})$, whereas some of them may be neutral to adversarial damage (e.g. a change between two wrong classes). The $\mathcal{L}_{0/1}$-adversarial damage thus lower bounds adversarial vulnerability. Both are even equal when the classifier is perfect (before attack), because then every change of label introduces an error. It is hence tempting to evaluate adversarial vulnerability with $\mathcal{L}_{0/1}$-adversarial damage.

**From $\Delta \mathcal{L}_{0/1}$ to $\Delta \mathcal{L}$ and to $\partial_{\boldsymbol{x}} \mathcal{L}$.** In practice however, we do not train our classifiers with the non-differentiable 0-1-loss but use a smoother loss $\mathcal{L}$, such as the cross-entropy loss. For similar reasons, we will now investigate the adversarial damage $\mathbb{E}_{\boldsymbol{x}}[\Delta \mathcal{L}(\boldsymbol{x}, c)]$ with loss $\mathcal{L}$ rather than $\mathcal{L}_{0/1}$. Like for Goodfellow et al. (2015); Lyu et al. (2015); Sinha et al. (2018) and many others, a classifier $\varphi$ will hence be robust if, on average over $\boldsymbol{x}$, a small adversarial perturbation $\boldsymbol{\delta}$ of $\boldsymbol{x}$ creates only a small variation $\delta \mathcal{L}$ of the loss. Now, if $\|\boldsymbol{\delta}\| \leq \epsilon$, then a first order Taylor expansion in $\epsilon$ shows that

$$\delta \mathcal{L} = \max_{\boldsymbol{\delta} : \|\boldsymbol{\delta}\| \leq \epsilon} |\mathcal{L}(\boldsymbol{x} + \boldsymbol{\delta}, c) - \mathcal{L}(\boldsymbol{x}, c)| \approx \max_{\boldsymbol{\delta} : \|\boldsymbol{\delta}\| \leq \epsilon} |\partial_{\boldsymbol{x}} \mathcal{L} \cdot \boldsymbol{\delta}| = \epsilon \, \|\|\partial_{\boldsymbol{x}} \mathcal{L}\|\|, \tag{2}$$

where $\partial_{\boldsymbol{x}} \mathcal{L}$ denotes the gradient of $\mathcal{L}$ with respect to $\boldsymbol{x}$, and where the last equality stems from the definition of the dual norm $\|\|\cdot\|\|$ of $\|\cdot\|$. Now two remarks. First: the dual norm only kicks in because we let the input noise $\boldsymbol{\delta}$ optimally adjust to the coordinates of $\partial_{\boldsymbol{x}} \mathcal{L}$ within its $\epsilon$-constraint. This is the brand mark of *adversarial* noise: the different coordinates add up, instead of statistically canceling each other out as they would with random noise. For example, if we impose that $\|\boldsymbol{\delta}\|_2 \leq \epsilon$, then $\boldsymbol{\delta}$ will strictly align with $\partial_{\boldsymbol{x}} \mathcal{L}$. If instead $\|\boldsymbol{\delta}\|_\infty \leq \epsilon$, then $\boldsymbol{\delta}$ will align with the sign of the coordinates of $\partial_{\boldsymbol{x}} \mathcal{L}$. Second remark: while the Taylor expansion in (2) becomes exact for infinitesimal perturbations, for finite ones it may actually be dominated by higher-order terms. Our experiments (Figures 1 & 2) however strongly suggest that in practice the first order term dominates the others. Now, remembering that the dual norm of an $\ell_p$-norm is the corresponding $\ell_q$-norm, and summarizing, we have proven

**Lemma 2.** *At first order approximation in $\epsilon$, an $\epsilon$-sized adversarial attack generated with norm $\|\cdot\|$ increases the loss $\mathcal{L}$ at point $\boldsymbol{x}$ by $\epsilon \|\|\partial_{\boldsymbol{x}} \mathcal{L}\|\|$, where $\|\|\cdot\|\|$ is the dual norm of $\|\cdot\|$. In particular, an $\epsilon$-sized $\ell_p$-attack increases the loss by $\epsilon \|\partial_{\boldsymbol{x}} \mathcal{L}\|_q$ where $1 \leq p \leq \infty$ and $\frac{1}{p} + \frac{1}{q} = 1$.*

Consequently, the adversarial damage of a classifier with loss $\mathcal{L}$ to $\epsilon$-sized attacks generated with norm $\|\cdot\|$ is $\epsilon \, \mathbb{E}_{\boldsymbol{x}} \|\|\partial_{\boldsymbol{x}} \mathcal{L}\|\|$. This is valid only at first order, but it proves that *at least* this kind of first-order vulnerability is present. We will see that the first-order predictions closely match the experiments, and that this insight helps protecting even against iterative (non-first-order) attack methods (Figure 1).

**Calibrating the threshold $\epsilon$ to the attack-norm $\|\cdot\|$.** Lemma 2 shows that adversarial vulnerability depends on three main factors: (i) $\|\cdot\|$, the norm chosen for the attack (ii) $\epsilon$, the size of the attack, and (iii) $\mathbb{E}_{\boldsymbol{x}} \|\|\partial_{\boldsymbol{x}} \mathcal{L}\|\|$, the expected *dual* norm of $\partial_{\boldsymbol{x}} \mathcal{L}$. We could see Point (i) as a measure of our sensibility to image perturbations, (ii) as our sensibility threshold, and (iii) as the classifier's expected marginal sensibility to a unit perturbation. $\mathbb{E}_{\boldsymbol{x}} \|\|\partial_{\boldsymbol{x}} \mathcal{L}\|\|$ hence intuitively captures the discrepancy between our perception (as modeled by $\|\cdot\|$) and the classifier's perception for an input-perturbation of small size $\epsilon$. Of course, this viewpoint supposes that we actually found a norm $\|\cdot\|$ (or more generally a metric) that faithfully reflects human perception – a project in its own right, far beyond the scope of this paper. However, it is clear that the threshold $\epsilon$ that we choose should depend on the norm $\|\cdot\|$ and hence on the input-dimension $d$. In particular, for a given pixel-wise order of magnitude of the perturbations $\boldsymbol{\delta}$, the $\ell_p$-norm of the perturbation will scale like $d^{1/p}$. This suggests to write the threshold $\epsilon_p$ used with $\ell_p$-attacks as:

$$\epsilon_p = \epsilon_\infty \, d^{1/p} , \tag{3}$$

where $\epsilon_\infty$ denotes a dimension-independent constant. In Appendix D we show that this scaling also preserves the average signal-to-noise ratio $\|\boldsymbol{x}\|_2 / \|\boldsymbol{\delta}\|_2$, both across norms and dimensions, so that $\epsilon_p$ could correspond to a constant human perception-threshold. With this in mind, the impatient reader may already jump to Section 3, which contains our main contributions: the estimation of $\mathbb{E}_x \|\boldsymbol{\partial}_{\boldsymbol{x}} \mathcal{L}\|_q$ for standard feed-forward nets. Meanwhile, the rest of this section shortly discusses two straightforward defenses that we will use later and that further illustrate the role of gradients.

**A new old regularizer.** Lemma 2 shows that the loss of the network after an $\frac{\epsilon}{2}$-sized $\|\cdot\|$-attack is

$$\mathcal{L}_{\epsilon, \|\cdot\|}(\boldsymbol{x}, c) := \mathcal{L}(\boldsymbol{x}, c) + \frac{\epsilon}{2} \|\!|\boldsymbol{\partial}_{\boldsymbol{x}} \mathcal{L}|\!\| . \tag{4}$$

It is thus natural to take this loss-after-attack as a new training objective. Here we introduced a factor 2 for reasons that will become clear in a moment. Incidentally, for $\|\cdot\| = \|\cdot\|_2$, this new loss reduces to an old regularization-scheme proposed by Drucker & LeCun (1991) called *double-backpropagation*. At the time, the authors argued that slightly decreasing a function's or a classifier's sensitivity to input perturbations should improve generalization. In a sense, this is exactly our motivation when defending against adversarial examples. It is thus not surprising to end up with the same regularization term. Note that our reasoning only shows that training with one specific norm $\|\!|\cdot|\!\|$ in (4) helps to protect against adversarial examples generated from $\|\cdot\|$. A priori, we do not know what will happen for attacks generated with other norms; but our experiments suggest that training with one norm also protects against other attacks (see Figure 2 and Section 4.1).

**Link to adversarially-augmented training.** In (1), $\epsilon$ designates an attack-size threshold, while in (4), it is a regularization-strength. Rather than a notation conflict, this reflects an intrinsic duality between two complementary interpretations of $\epsilon$, which we now investigate further. Suppose that, instead of using the loss-after-attack, we augment our training set with $\epsilon$-sized $\|\cdot\|$-attacks $\boldsymbol{x} + \boldsymbol{\delta}$, where for each training point $\boldsymbol{x}$, the perturbation $\boldsymbol{\delta}$ is generated on the fly to locally maximize the loss-increase. Then we are effectively training with

$$\tilde{\mathcal{L}}_{\epsilon, \|\cdot\|}(\boldsymbol{x}, c) := \frac{1}{2}(\mathcal{L}(\boldsymbol{x}, c) + \mathcal{L}(\boldsymbol{x} + \epsilon \boldsymbol{\delta}, c)) , \tag{5}$$

where by construction $\boldsymbol{\delta}$ satisfies (2). We will refer to this technique as *adversarially augmented training*. It was first introduced by Goodfellow et al. (2015) with $\|\cdot\| = \|\cdot\|_\infty$ under the name of FGSM[1]-augmented training. Using the first order Taylor expansion in $\epsilon$ of (2), this 'old-plus-post-attack' loss of (5) simply reduces to our loss-after-attack, which proves

**Proposition 3.** *Up to first-order approximations in $\epsilon$, $\tilde{\mathcal{L}}_{\epsilon, \|\cdot\|} = \mathcal{L}_{\epsilon, \|\!|\cdot|\!\|}$ . Said differently, for small enough $\epsilon$, adversarially-augmented training with $\epsilon$-sized $\|\cdot\|$-attacks amounts to penalizing the* dual *norm $\|\!|\cdot|\!\|$ of $\boldsymbol{\partial}_{\boldsymbol{x}} \mathcal{L}$ with weight $\epsilon/2$. In particular, double-backpropagation corresponds to training with $\ell_2$-attacks, while FGSM-augmented training corresponds to an $\ell_1$-penalty on $\boldsymbol{\partial}_{\boldsymbol{x}} \mathcal{L}$.*

This correspondence between training with perturbations and using a regularizer can be compared to Tikhonov regularization: Tikhonov regularization amounts to training with *random* noise Bishop (1995), while training with *adversarial* noise amounts to penalizing $\boldsymbol{\partial}_{\boldsymbol{x}} \mathcal{L}$. Section 4.1 verifies the correspondence between adversarial augmentation and gradient regularization empirically, which also strongly suggests the empirical validity of the first-order Taylor expansion in (2).

## 3 ESTIMATING $\|\boldsymbol{\partial}_{\boldsymbol{x}} \mathcal{L}\|_q$ TO EVALUATE ADVERSARIAL VULNERABILITY

In this section, we evaluate the size of $\|\boldsymbol{\partial}_{\boldsymbol{x}} \mathcal{L}\|_q$ for standard neural network architectures. We start with fully-connected networks, and finish with a much more general theorem that, not only encompasses CNNs (with or without strided convolutions), but also shows that the gradient-norms are essentially independent of the network topology. We start our analysis by showing how changing $q$ affects the size of $\|\boldsymbol{\partial}_{\boldsymbol{x}} \mathcal{L}\|_q$. Suppose for a moment that the coordinates of $\boldsymbol{\partial}_{\boldsymbol{x}} \mathcal{L}$ have typical magnitude $|\boldsymbol{\partial}_{\boldsymbol{x}} \mathcal{L}|$. Then $\|\boldsymbol{\partial}_{\boldsymbol{x}} \mathcal{L}\|_q$ scales like $d^{1/q}|\boldsymbol{\partial}_{\boldsymbol{x}} \mathcal{L}|$. Consequently

$$\epsilon_p \|\boldsymbol{\partial}_{\boldsymbol{x}} \mathcal{L}\|_q \; \propto \; \epsilon_p \, d^{1/q} |\boldsymbol{\partial}_{\boldsymbol{x}} \mathcal{L}| \; \propto \; d \, |\boldsymbol{\partial}_{\boldsymbol{x}} \mathcal{L}| . \tag{6}$$

---

[1]FGSM = *F*ast *G*radient *S*ign *M*ethod

This equation carries two important messages. First, we see how $\|\boldsymbol{\partial}_x \mathcal{L}\|_q$ depends on $d$ and $q$. The dependence seems highest for $q = 1$. But once we account for the varying perceptibility threshold $\epsilon_p \propto d^{1/p}$, we see that adversarial vulnerability scales like $d \cdot |\boldsymbol{\partial}_x \mathcal{L}|$, whatever $\ell_p$-norm we use. Second, (6) shows that to be robust against any type of $\ell_p$-attack at any input-dimension $d$, the average absolute value of the coefficients of $\boldsymbol{\partial}_x \mathcal{L}$ must grow slower than $1/d$. Now, here is the catch, which brings us to our core insight.

### 3.1 CORE IDEA: ONE NEURON WITH MANY INPUTS

In order to preserve the activation variance of the neurons from layer to layer, the neural weights are usually initialized with a variance that is inversely proportional to the number of inputs per neuron. Imagine for a moment that the network consisted only of one output neuron $o$ linearly connected to all input pixels. For the purpose of this example, we assimilate $o$ and $\mathcal{L}$. Because we initialize the weights with a variance of $1/d$, their average absolute value $|\boldsymbol{\partial}_x o| \equiv |\boldsymbol{\partial}_x \mathcal{L}|$ grows like $1/\sqrt{d}$, rather than the required $1/d$. By (6), the adversarial vulnerability $\epsilon \|\boldsymbol{\partial}_x o\|_q \equiv \epsilon \|\boldsymbol{\partial}_x \mathcal{L}\|_q$ therefore increases like $d/\sqrt{d} = \sqrt{d}$.

*This toy example shows that the standard initialization scheme, which preserves the variance from layer to layer, causes the average coordinate-size $|\boldsymbol{\partial}_x \mathcal{L}|$ to grow like $1/\sqrt{d}$ instead of $1/d$. When an $\ell_\infty$-attack tweaks its $\epsilon$-sized input-perturbations to align with the coordinate-signs of $\boldsymbol{\partial}_x \mathcal{L}$, all coordinates of $\boldsymbol{\partial}_x \mathcal{L}$ add up in absolute value, resulting in an output-perturbation that scales like $\epsilon\sqrt{d}$ and leaves the network increasingly vulnerable with growing input-dimension.*

### 3.2 GENERALIZATION TO DEEP NETWORKS

Our next theorems generalize the previous toy example to a very wide class of feedforward nets with ReLU activation functions. For illustration purposes, we start with fully connected nets and only then proceed to the broader class, which includes any succession of (possibly strided) convolutional layers. In essence, the proofs iterate our insight on one layer over a sequence of layers. They all rely on the following set ($\mathcal{H}$) of hypotheses:

H1 Non-input neurons are followed by a ReLU killing half of its inputs, independently of the weights.
H2 Neurons are partitioned into layers, meaning groups that each path traverses at most once.
H3 All weights have 0 expectation and variance $2/$(in-degree) ('He-initialization').
H4 The weights from different layers are independent.
H5 Two distinct weights $w, w'$ from a same node satisfy $\mathbb{E}\left[w\, w'\right] = 0$.

If we follow common practice and initialize our nets as proposed by He et al. (2015), then H3-H5 are satisfied at initialization by design, while H1 is usually a very good approximation (Balduzzi et al., 2017). Note that such i.i.d. weight assumptions have been widely used to analyze neural nets and are at the heart of very influential and successful prior work (e.g., equivalence between neural nets and Gaussian processes as pioneered by Neal 1996). Nevertheless, they do not hold after training. That is why all our statements in this section are to be understood as *orders of magnitudes* that are very well satisfied at initialization in theory and in practice, and that we will confirm experimentally after training in Section 4. Said differently, while our theorems rely on the statistics of neural nets at initialization, our experiments confirm their conclusions after training.

**Theorem 4 (Vulnerability of Fully Connected Nets).** *Consider a succession of fully connected layers with ReLU activations which takes inputs $\boldsymbol{x}$ of dimension $d$, satisfies assumptions ($\mathcal{H}$), and outputs logits $f_k(\boldsymbol{x})$ that get fed to a final cross-entropy-loss layer $\mathcal{L}$. Then the coordinates of $\boldsymbol{\partial}_x f_k$ grow like $1/\sqrt{d}$, and*

$$\|\boldsymbol{\partial}_x \mathcal{L}\|_q \propto d^{\frac{1}{q}-\frac{1}{2}} \quad \text{and} \quad \epsilon_p \|\boldsymbol{\partial}_x \mathcal{L}\|_q \propto \sqrt{d}. \tag{7}$$

*These networks are thus increasingly vulnerable to $\ell_p$-attacks with growing input-dimension.*

Theorem 4 is a special case of the next theorem, which will show that the previous conclusions are essentially independent of the network-topology. We will use the following symmetry assumption on the neural connections. For a given path $\boldsymbol{p}$, let the *path-degree* $d_{\boldsymbol{p}}$ be the multiset of encountered in-degrees along path $\boldsymbol{p}$. For a fully connected network, this is the unordered sequence of layer-sizes

preceding the last path-node, including the input-layer. Now consider the multiset $\{d_{\boldsymbol{p}}\}_{\boldsymbol{p}\in\mathcal{P}(x,o)}$ of all path-degrees when $\boldsymbol{p}$ varies among all paths from input $x$ to output $o$. The symmetry assumption (relatively to $o$) is

$\quad$ ($\mathcal{S}$) All input nodes $x$ have the same multiset $\{d_{\boldsymbol{p}}\}_{\boldsymbol{p}\in\mathcal{P}(x,o)}$ of path-degrees from $x$ to $o$.

Intuitively, this means that the statistics of degrees encountered along paths to the output are the same for all input nodes. This symmetry assumption is exactly satisfied by fully connected nets, almost satisfied by CNNs (up to boundary effects, which can be alleviated via periodic or mirror padding) and exactly satisfied by strided layers, if the layer-size is a multiple of the stride.

**Theorem 5** (**Vulnerability of Feedforward Nets**). *Consider any feed-forward network with linear connections and ReLU activation functions. Assume the net satisfies assumptions ($\mathcal{H}$) and outputs logits $f_k(\boldsymbol{x})$ that get fed to the cross-entropy-loss $\mathcal{L}$. Then $\|\partial_{\boldsymbol{x}} f_k\|_2$ is independent of the input dimension $d$ and $\epsilon_2 \|\partial_{\boldsymbol{x}}\mathcal{L}\|_2 \propto \sqrt{d}$. Moreover, if the net satisfies the symmetry assumption ($\mathcal{S}$), then $|\partial_x f_k| \propto 1/\sqrt{d}$ and (7) still holds: $\|\partial_{\boldsymbol{x}}\mathcal{L}\|_q \propto d^{\frac{1}{q}-\frac{1}{2}}$ and $\epsilon_p \|\partial_{\boldsymbol{x}}\mathcal{L}\|_q \propto \sqrt{d}$.*

Theorems 4 and 5 are proven in Appendix B. The main proof idea is that in the gradient norm computation, the He-initialization exactly compensates the combinatorics of the number of paths in the network, so that this norm becomes independent of the network topology. In particular, we get

**Corollary 6** (**Vulnerability of CNNs**). *In any succession of convolution and dense layers, strided or not, with ReLU activations, that satisfies assumptions ($\mathcal{H}$) and outputs logits that get fed to the cross-entropy-loss $\mathcal{L}$, the gradient of the logit-coordinates scale like $1/\sqrt{d}$ and (7) is satisfied. It is hence increasingly vulnerable with growing input-resolution to attacks generated with any $\ell_p$-norm.*

Appendix A shows that the network gradient are dampened when replacing strided layers by average poolings, essentially because average-pooling weights do not follow the He-init assumption H3.

## 4 Empirical Results

In Section 4.1, we empirically verify the validity of the first-order Taylor approximation made in (2) (Fig.1), for example by checking the correspondence between loss-gradient regularization and adversarially-augmented training (Fig.2). Section 4.2 then empirically verifies that both the average $\ell_1$-norm of $\partial_{\boldsymbol{x}}\mathcal{L}$ and the adversarial vulnerability grow like $\sqrt{d}$ as predicted by Corollary 6. For all experiments, we approximate adversarial vulnerability using various attacks of the Foolbox-package (Rauber et al., 2017). We use an $\ell_\infty$ attack-threshold of size $\epsilon_\infty = 0.005$ (and later $0.002$) which, for pixel-values ranging from 0 to 1, is completely imperceptible but suffices to fool the classifiers on a significant proportion of examples. This $\epsilon_\infty$-*threshold* should not be confused with the *regularization-strengths* $\epsilon$ appearing in (4) and (5), which will be varied in some experiments.

### 4.1 First-Order Approximation, Gradient Penalty, Adversarial Augmentation

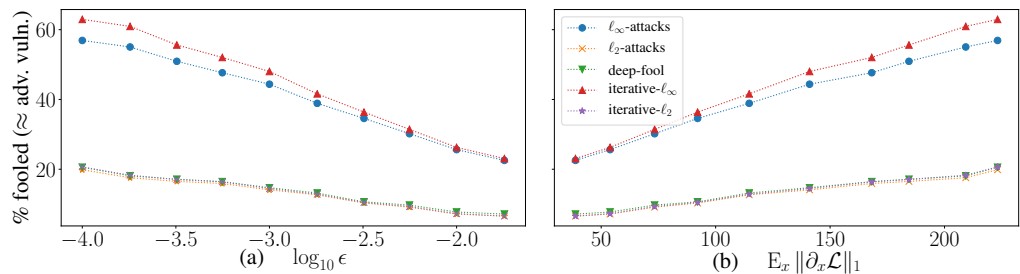

Figure 1: Adversarial vulnerability approximated by different attack-types for 10 trained networks as a function of ($a$) the $\ell_1$ gradient regularization-strength $\epsilon$ used to train the nets and ($b$) the average gradient-norm. These curves confirm that the first-order expansion term in (2) is a crucial component of adversarial vulnerability.

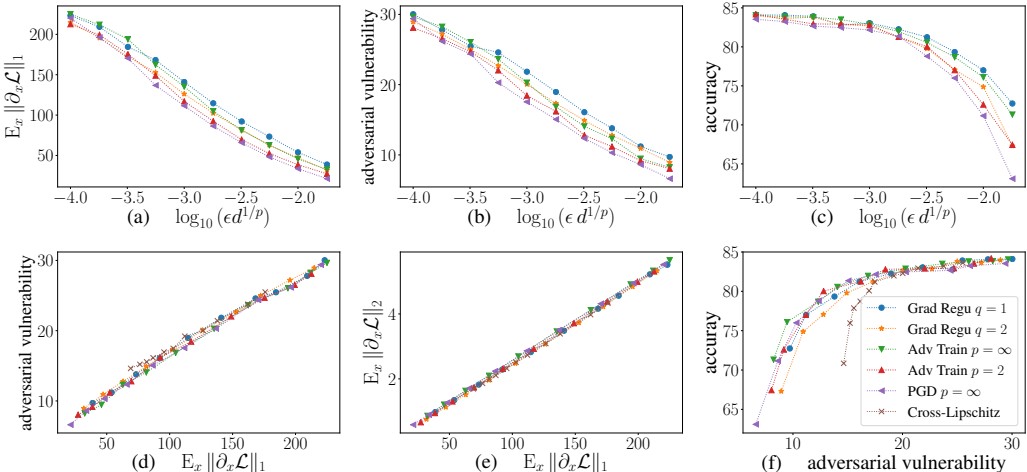

Figure 2: Average norm $\mathbb{E}_{\boldsymbol{x}}\|\boldsymbol{\partial}_{\boldsymbol{x}}\mathcal{L}\|$ of the loss-gradients, adversarial vulnerability and accuracy (before attack) of various networks trained with different adversarial regularization methods and regularization strengths $\epsilon$. Each point represents a trained network, and each curve a training-method. *Upper row*: A priori, the regularization-strengths $\epsilon$ have different meanings for each method. The near superposition of all upper-row curves illustrates $(i)$ the duality between adversarial augmentation and gradient-regularization (Prop. 3) and $(ii)$ confirms the rescaling of $\epsilon$ proposed in (3) and $(iii)$ supports the validity of the first-order Taylor expansion (2). $(d)$: near functional relation between adversarial vulnerability and average loss-gradient norms. $(e)$: the near-perfect linear relation between the $\mathbb{E}\|\boldsymbol{\partial}_{\boldsymbol{x}}\mathcal{L}\|_1$ and $\mathbb{E}\|\boldsymbol{\partial}_{\boldsymbol{x}}\mathcal{L}\|_2$ suggests that protecting against a given attack-norm also protects against others. $(f)$: Merging 2band 2c shows that all adversarial augmentation and gradient-regularization methods achieve similar accuracy-vulnerability trade-offs.

We train several CNNs with same architecture to classify CIFAR-10 images (Krizhevsky, 2009). For each net, we use a specific training method with a specific regularization value $\epsilon$. The training methods used were $\ell_1$- and $\ell_2$-penalization of $\boldsymbol{\partial}_{\boldsymbol{x}}\mathcal{L}$ (Eq. 4), adversarial augmentation with $\ell_\infty$- and $\ell_2$- attacks (Eq. 5), projected gradient descent (PGD) with randomized starts (7 steps per attack with step-size = $.2\,\epsilon_\infty$; see Madrix et al. 2018) and the cross-Lipschitz regularizer (Eq. 17 in Appendix C). We then test the adversarial vulnerability of each trained network using the following attack-methods: single-step $\ell_\infty$- (FGSM) and $\ell_2$-attacks, iterative $\ell_\infty$- (PGD) and $\ell_2$-attacks, and DeepFool attacks (Moosavi-Dezfooli et al., 2016). All networks have 6 'strided convolution $\rightarrow$ batchnorm $\rightarrow$ ReLU' layers with strides [1, 2, 2, 2, 2, 2] respectively and 64 output-channels each, followed by a final fully-connected linear layer. Results are summarized in Figures 1 and 2. Figure 1 fixes the training method – gradient $\ell_1$-regularization – and plots the obtained adversarial vulnerabilities for various attacks types. Figure 2 fixes the attack type – iterative $\ell_\infty$-attacks – but plots the curves obtained for various training methods. Note that our goal here is not to advocate one defense over another, but rather to check the validity of the Taylor expansion, and empirically verify that first order terms (i.e., gradients) suffice to explain much of the observed adversarial vulnerability. Similarly, our goal in testing several attacks (Figure 1) is not to present a specifically strong one, but rather to verify that for all attacks, the trends are the same: the vulnerability grows with increasing gradients.

**Validity of first order expansion.** The following observations support the validity of the first order Taylor expansion in (2) and suggest that it is a crucial component of adversarial vulnerability: (i) the efficiency of the first-order defense against iterative (non-first-order) attacks (Fig.1a); (ii) the striking similarity between the PGD curves (adversarial augmentation with *iterative* attacks) and the other adversarial training training curves (*one-step* attacks/defenses); (iii) the functional-like dependence between any approximation of adversarial vulnerability and $\mathbb{E}_{\boldsymbol{x}}\|\boldsymbol{\partial}_{\boldsymbol{x}}\mathcal{L}\|_1$ (Fig.1b), and its independence on the training method (Fig.2d). (iv) the excellent correspondence between the gradient-regularization and adversarial training curves (see next paragraph). Said differently, adversarial examples seem indeed to be primarily caused by large gradients of the classifier as captured via the induced loss. [2]

---

[2]On Figure 1, the two $\ell_\infty$-attacks seem more efficient than the others, because we chose an $\ell_\infty$ perturbation threshold ($\epsilon_\infty$). With an $\ell_2$-threshold it is the opposite (see Figure 7, Appendix F).

**Illustration of Proposition 3.** The upper row of Figure 2 plots $\mathbb{E}_{\boldsymbol{x}}\|\partial_{\boldsymbol{x}}\mathcal{L}_1\|$, adversarial vulnerability and accuracy as a function of $\epsilon\, d^{1/p}$. The excellent match between the adversarial augmentation curve with $p = \infty$ ($p = 2$) and its gradient-regularization dual counterpart with $q = 1$ (resp. $q = 2$) illustrates the duality between $\epsilon$ as a threshold for adversarially-augmented training and as a regularization constant in the regularized loss (Proposition 3). *It also supports the validity of the first-order Taylor expansion in* (2).

**Confirmation of** (3). Still on the upper row, the curves for $p = \infty, q = 1$ have no reason to match those for $p = q = 2$ when plotted against $\epsilon$, because $\epsilon$-threshold is relative to a specific attack-norm. However, (3) suggested that the rescaled thresholds $\epsilon d^{1/p}$ may approximately correspond to a same 'threshold-unit' across $\ell_p$-norms and across dimension. This is well confirmed by the upper row plots: by rescaling the x-axis, the $p = q = 2$ and $q = 1, p = \infty$ curves get almost super-imposed.

**Accuracy-vs-Vulnerability Trade-Off.** Merging Figures 2b and 2c by taking out $\epsilon$, Figure 2f shows that all gradient regularization and adversarial training methods yield equivalent accuracy-vulnerability trade-offs. Incidentally, for higher penalization values, these trade-offs appear to be much better than those given by cross Lipschitz regularization.

**The penalty-norm does not matter.** We were surprised to see that on Figures 2d and 2f, the $\mathcal{L}_{\epsilon,q}$ curves are almost identical for $q = 1$ and $2$. This indicates that both norms can be used interchangeably in (4) (modulo proper rescaling of $\epsilon$ via (3)), and suggests that protecting against a specific attack-norm also protects against others. (6) may provide an explanation: if the coordinates of $\partial_{\boldsymbol{x}}\mathcal{L}$ behave like centered, uncorrelated variables with equal variance –which follows from assumptions $(\mathcal{H})$ –, then the $\ell_1$- and $\ell_2$-norms of $\partial_{\boldsymbol{x}}\mathcal{L}$ are simply proportional. Plotting $\mathbb{E}_{\boldsymbol{x}}\|\partial_{\boldsymbol{x}}\mathcal{L}(\boldsymbol{x})\|_2$ against $\mathbb{E}_{\boldsymbol{x}}\|\partial_{\boldsymbol{x}}\mathcal{L}(\boldsymbol{x})\|_1$ in Figure 2e confirms this explanation. The slope is independent of the training method. Therefore, penalizing $\|\partial_{\boldsymbol{x}}\mathcal{L}(\boldsymbol{x})\|_1$ during training will not only decrease $\mathbb{E}_{\boldsymbol{x}}\|\partial_{\boldsymbol{x}}\mathcal{L}\|_1$ (as shown in Figure 2a), but also drive down $\mathbb{E}_{\boldsymbol{x}}\|\partial_{\boldsymbol{x}}\mathcal{L}\|_2$ and vice-versa.

## 4.2 VULNERABILITY GROWS WITH INPUT RESOLUTION

Theorems 4-5 and Corollary 6 predict a linear growth of the average $\ell_1$-norm of $\partial_{\boldsymbol{x}}\mathcal{L}$ with the square root of the input dimension $d$, and therefore also of adversarial vulnerability (Lemma 2). To test these predictions, we upsampled the CIFAR-10 images (of size 3 x 32 x 32) by copying pixels so as to get 4 datasets with, respectively, 32, 64, 128 and 256 pixels per edge. We then trained a CNN on each dataset

and computed their adversarial vulnerability (with iterative $\ell_\infty$-attacks, threshold $\epsilon_\infty = .002$) and average $\|\partial_{\boldsymbol{x}}\mathcal{L}\|_1$ over the last 20 epochs on the same held-out test-dataset. This gave us 2 x 20-values per net and image-size, summarized in Figure 3. The dashed-lines follow their medians and the errorbars show their $10^{\text{th}}$ and $90^{\text{th}}$ quantiles. As predicted by our theorems, both $\|\partial_{\boldsymbol{x}}\mathcal{L}\|_1$ and adversarial vulnerability grow approximately linearly with $\sqrt{d}$. We also ran a similar experiment on downsized ImageNet images, where we train several identical nets per image-size rather than just one. Conclusions are unchanged. See Appendix E.

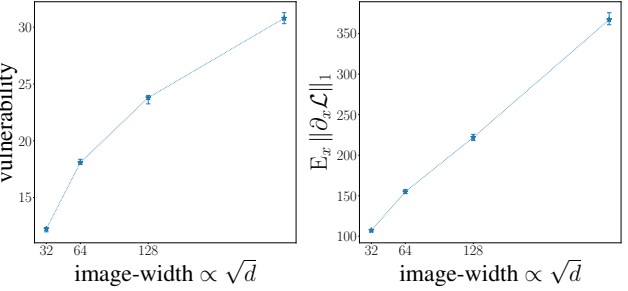

Figure 3: $\mathbb{E}_{\boldsymbol{x}}\|\partial_{\boldsymbol{x}}\mathcal{L}\|_1$ (right) increase linearly with the square-root of the image-resolution $d$, as predicted by Corollary 6. Adversarial vulnerability (left) increases similarly with $d$, but gets slightly dampened with increasing dimension, probably because the first-order approximation made in (2) becomes less and less valid.

All networks had exactly the same amount of parameters and very similar structure across the various input-resolutions. The CNNs were a succession of 8 'convolution $\rightarrow$ batchnorm $\rightarrow$ ReLU' layers with 64 output channels, followed by a final full-connection to the 12 logit-outputs. We used $2 \times 2$-max-poolings after the convolutions of layers 2, 4, 6 and 8, and a final max-pooling after layer 8 that fed only 1 neuron per channel to the fully-connected layer. To ensure that the convolution-kernels cover similar ranges of the images across each of the 32, 64, 128 and 256 input-resolutions, we respectively dilated all convolutions ('à trous') by a factor 1, 2, 4 and 8.

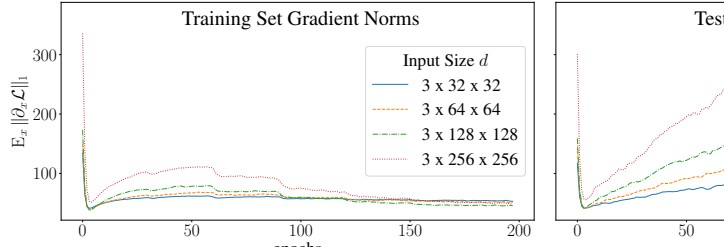

Figure 4: Evolution over training epochs of the average $\ell_1$-gradient-norms on the training (left) and test set (right). There is a clear discrepancy: on the training set, the gradient norms decrease (after an initialization phase) and are dimension-independent; on the test set, they increase and scale like $\sqrt{d}$. This suggests that, outside the training points, the nets tend to recover their prior gradient-properties (i.e. naturally large gradients).

## 5 DISCUSSIONS

### 5.1 IMPLICATIONS: WHY PRIOR VULNERABILITY MAY MATTER

Our theoretical results show that the priors of classical neural networks yield vulnerable functions because of naturally high gradients. And our experiments (Fig 3&6) suggest that usual training does not escape these prior properties. But how may these insights help understanding the vulnerability of robustly trained networks? Clearly, to be successful, robust training algorithms must escape ill-behaved priors, which explains why most methods (e.g. FGSM, PGD) are essentially gradient penalization techniques. But, MNIST aside, even state-of-the-art methods largely fail at protecting current network architectures (Madry et al., 2018), and understanding why is motivation to this and many other papers. Interestingly, Schmidt et al. (2018) recently noticed that those methods actually do protect the nets on training examples, but fail to generalize to the test set. They hence conclude that state-of-the-art robustification algorithms work, but need more data. Alternatively however, when generalization fails, one can also reduce the model's complexity. Large fully connected nets for example typically fail to generalize to out-of-sample examples: getting similar accuracies than CNNs would need prohibitively many training points. Similarly, Schmidt et al.'s observations may suggest that, outside the training points, networks tend to recover their prior properties, i.e. naturally large gradients. Figure 4 corroborates this hypothesis. It plots the evolution over training epochs of the $\ell_1$-gradient-norms of the CNNs from Section 4.2 (Fig 3) on the training and test sets respectively. The discrepancy is unmistakable: after a brief initialization phase, the norms decrease on the training set, but increase on the test set. They are moreover almost input-dimension independent on the training set, but scale as $\sqrt{d}$ on the test set (as seen in Fig 3) up to respectively 2, 4, 8 and 16 times the training set values. These observations suggest that, with the current amount of data, tackling adversarial vulnerability may require new architectures with inherently smaller gradients. Searching these architectures among those with well-behaved prior-gradients seems a reasonable start, where our theoretical results may prove very useful.[3]

### 5.2 RELATED LITERATURE

**On network vulnerability.** Goodfellow et al. (2015) already stressed that adversarial vulnerability increases with growing dimension $d$. But their argument only relied on a *linear* 'one-output-to-many-inputs'-model with *dimension-independent* weights. They therefore concluded on a linear growth of adversarial vulnerability with $d$. In contrast, our theory applies to almost any standard feed-forward architecture (not just linear), and shows that, once we adjust for the weight's dimension-dependence, adversarial vulnerability increases like $\sqrt{d}$ (not $d$), *almost independently of the architecture*. Nevertheless, our experiments confirm Goodfellow et al.'s idea that our networks are "too linear-like", in the sense that a first-order Taylor expansion is indeed sufficient to explain the adversarial vulnerability of neural networks. As suggested by the one-output-to-many-inputs model, the culprit is that growing

---

[3]Appendix A investigates such a preliminary direction by introducing average poolings, which have a weight-size $1/\text{in-channels}$ rather than the typical $1/\sqrt{\text{in-channels}}$ of the other He-initialized weights.

dimensionality gives the adversary more and more room to 'wriggle around' with the noise and adjust to the gradient of the output neuron. This wriggling, we show, is still possible when the output is connected to all inputs only indirectly, even when no neuron is directly connected to all inputs, like in CNNs. This explanation of adversarial vulnerability is independent of the *intrinsic* dimensionality or geometry of the data (compare to Amsaleg et al. 2017; Gilmer et al. 2018). Finally, let us mention that Fawzi et al. (2016) show a close link between the vulnerability to small worst-case perturbation (as studied here) and larger average perturbations. Our findings on the adversarial vulnerability NNs to small perturbation could thus be translated accordingly.

**On robustification algorithms.** Incidentally, Goodfellow et al. (2015) also already relate adversarial vulnerability to large gradients of the loss $\mathcal{L}$, an insight at the very heart of their FGSM-algorithm. They however do not propose any explicit penalizer on the gradient of $\mathcal{L}$ other than indirectly through adversarially-augmented training. Conversely, Ross & Doshi-Velez (2018) propose the old double-backpropagation to robustify networks but make no connection to FGSM and adversarial augmentation. Lyu et al. (2015) discuss and use the connection between gradient-penalties and adversarial augmentation, but never actually compare both in experiments. This comparison however is essential to test the validity of the first-order Taylor expansion in (2), as confirmed by the similarity between the gradient-regularization and adversarial-augmentation curves in Figure 2. Hein & Andriushchenko (2017) derived yet another gradient-based penalty –the *cross-Lipschitz*-penalty– by considering (and proving) formal guarantees on adversarial vulnerability itself, rather than adversarial damage. While both penalties are similar in spirit, focusing on the adversarial damage rather than vulnerability has two main advantages. First, it achieves better accuracy-to-vulnerability ratios, both in theory and practice, because it ignores class-switches between misclassified examples and penalizes only those that reduce the accuracy. Second, it allows to deal with one number only, $\Delta\mathcal{L}$, whereas Hein & Andriushchenko's cross-Lipschitz regularizer and theoretical guarantees explicitly involve *all* $K$ logit-functions (and their gradients). See Appendix C. Penalizing network-gradients is also at the heart of contractive auto-encoders as proposed by Rifai et al. (2011), where it is used to regularize the encoder-features. Seeing adversarial training as a generalization method, let us also mention Hochreiter & Schmidhuber (1995), who propose to enhance generalization by searching for parameters in a "flat minimum region" of the loss. This leads to a penalty involving the gradient of the loss, but taken with respect to the weights, rather than the inputs. In the same vein, a gradient-regularization of the loss of generative models also appears in Proposition 6 of Ollivier (2014), where it stems from a code-length bound on the data (minimum description length). More generally, the gradient regularized objective (4) is essentially the first-order approximation of the robust training objective $\max_{\|\delta\|\le\epsilon} \mathcal{L}(x + \delta, c)$ which has a long history in math (Wald, 1945), machine learning (Xu et al., 2009) and now adversarial vulnerability (Sinha et al., 2018). Finally, Cisse et al. (2017) propose new network-architectures that have small gradients by design, rather than by special training: an approach that makes all the more sense, considering the conclusion of Theorems 4 and 5. For further details and references on adversarial attacks and defenses, we refer to Yuan et al. (2017).

# 6 CONCLUSION

For differentiable classifiers and losses, we showed that adversarial vulnerability increases with the gradients $\partial_x\mathcal{L}$ of the loss, which is confirmed by the near-perfect functional relationship between gradient norms and vulnerability (Figures 1&2d). We then evaluated the size of $\|\partial_x\mathcal{L}\|_q$ and showed that, at initialization, usual feed-forward nets (convolutional or fully connected) are increasingly vulnerable to $\ell_p$-attacks with growing input dimension $d$ (the image-size), almost independently of their architecture. Our experiments show that, on the tested architectures, usual training escapes those prior gradient (and vulnerability) properties on the training, but not on the test set. Schmidt et al. (2018) suggest that alleviating this generalization gap requires more data. But a natural (complementary) alternative would be to search for architectures with naturally smaller gradients, and in particular, with well-behaved priors. Despite all their limitations (being only first-order, assuming a prior weight-distribution and a differentiable loss and architecture), our theoretical insights may thereby still prove to be precious future allies.

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

## A  EFFECTS OF STRIDED AND AVERAGE-POOLING LAYERS ON ADVERSARIAL VULNERABILITY

It is common practice in CNNs to use average-pooling layers or strided convolutions to progressively decrease the number of pixels per channel. Corollary 6 shows that using strided convolutions does not protect against adversarial examples. However, what if we replace strided convolutions by convolutions with stride 1 plus an average-pooling layer? Theorem 5 considers only *randomly* initialized weights with typical size $1/\sqrt{\text{in-degree}}$. Average-poolings however introduce *deterministic* weights of size $1/(\text{in-degree})$. These are smaller and may therefore dampen the input-to-output gradients and protect against adversarial examples. We confirm this in our next theorem, which uses a slightly modified version $(\mathcal{H}')$ of $(\mathcal{H})$ to allow average pooling layers. $(\mathcal{H}')$ is $(\mathcal{H})$, but where the He-init H3 applies to all weights *except* the (deterministic) average pooling weights, and where H1 places a ReLU on every non-input *and non-average-pooling* neuron.

**Theorem 7** (**Effect of Average-Poolings**). *Consider a succession of convolution layers, dense layers and $n$ average-pooling layers, in any order, that satisfies $(\mathcal{H}')$ and outputs logits $f_k(\boldsymbol{x})$. Assume the $n$ average pooling layers have a stride equal to their mask size and perform averages over $a_1$, ..., $a_n$ nodes respectively. Then $\|\partial_{\boldsymbol{x}} f_k\|_2$ and $|\partial_x f_k|$ scale like $1/\sqrt{a_1 \cdots a_n}$ and $1/\sqrt{d\, a_1 \cdots a_n}$ respectively.*

Proof in Appendix B.4. Theorem 7 suggest to try and replace any strided convolution by its non-strided counterpart, followed by an average-pooling layer. It also shows that if we systematically reduce the number of pixels per channel down to 1 by using only non-strided convolutions and average-pooling layers (i.e. $d = \prod_{i=1}^{n} a_i$), then all input-to-output gradients should become independent of $d$, thereby making the network completely robust to adversarial examples.

Our following experiments (Figure 5) show that after training, the networks get indeed robustified to adversarial examples, but remain more vulnerable than suggested by Theorem 7.

**Experimental setup.** Theorem 7 shows that, contrary to strided layers, average-poolings should decrease adversarial vulnerability. We tested this hypothesis on CNNs trained on CIFAR-10, with 6 blocks of 'convolution $\to$ BatchNorm $\to$ ReLU' with 64 output-channels, followed by a final average pooling feeding one neuron per channel to the last fully-connected linear layer. Additionally, after every second convolution, we placed a pooling layer with stride and mask-size $(2, 2)$ (thus acting on $2 \times 2$

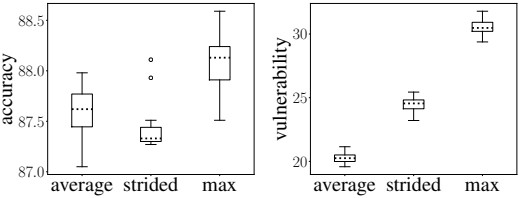

Figure 5: As predicted by Theorem 7, average-pooling layers make networks more robust to adversarial examples, contrary to strided (and max-pooling) ones. But the vulnerability with average-poolings remains higher than anticipated.

neurons at a time, without overlap). We tested average-pooling, strided and max-pooling layers and trained 20 networks per architecture. Results are shown in Figure 5. All accuracies are very close, but, as predicted, the networks with average pooling layers are more robust to adversarial images than the others. However, they remain more vulnerable than what would follow from Theorem 7. We also noticed that, contrary to the strided architectures, their gradients after training are an order of magnitude higher than at initialization and than predicted. This suggests that assumptions $(\mathcal{H})$ get more violated when using average-poolings instead of strided layers. Understanding why will need further investigations.

## B  PROOFS

### B.1  PROOF OF PROPOSITION 3

*Proof.* Let $\epsilon\,\boldsymbol{\delta}$ be an adversarial perturbation with $\|\boldsymbol{\delta}\| = 1$ that locally maximizes the loss increase at point $\boldsymbol{x}$, meaning that $\boldsymbol{\delta} = \arg\max_{\|\boldsymbol{\delta}'\| \leq 1} \partial_{\boldsymbol{x}}\mathcal{L} \cdot \boldsymbol{\delta}'$. Then, by definition of the dual norm of $\partial_{\boldsymbol{x}}\mathcal{L}$ we have: $\partial_{\boldsymbol{x}}\mathcal{L} \cdot (\epsilon\boldsymbol{\delta}) = \epsilon \|\partial_{\boldsymbol{x}}\mathcal{L}\|$. Thus

$$\tilde{\mathcal{L}}_{\epsilon,\|\cdot\|}(\boldsymbol{x},c) = \frac{1}{2}(\mathcal{L}(\boldsymbol{x},c) + \mathcal{L}(\boldsymbol{x}+\epsilon\,\boldsymbol{\delta},c)) = \frac{1}{2}(2\mathcal{L}(\boldsymbol{x},c) + \epsilon\,|\boldsymbol{\partial}_{\boldsymbol{x}}\mathcal{L}\cdot\boldsymbol{\delta}| + o(\|\boldsymbol{\delta}\|)) =$$

$$= \mathcal{L}(\boldsymbol{x},c) + \frac{\epsilon}{2}\,\|\boldsymbol{\partial}_{\boldsymbol{x}}\mathcal{L}\| + o(\epsilon) = \mathcal{L}_{\epsilon,\|\cdot\|}(\boldsymbol{x},c) + o(\epsilon)\,. \quad \square$$

## B.2 PROOF OF THEOREM 4

*Proof.* Let $x$ designate a generic coordinate of $\boldsymbol{x}$. To evaluate the size of $\|\boldsymbol{\partial}_{\boldsymbol{x}}\mathcal{L}\|_q$, we will evaluate the size of the coordinates $\partial_x\mathcal{L}$ of $\boldsymbol{\partial}_{\boldsymbol{x}}\mathcal{L}$ by decomposing them into

$$\partial_x\mathcal{L} = \sum_{k=1}^{K} \frac{\partial\mathcal{L}}{\partial f_k}\frac{\partial f_k}{\partial x} =: \sum_{k=1}^{K} \partial_k\mathcal{L}\,\partial_x f_k,$$

where $f_k(\boldsymbol{x})$ denotes the logit-probability of $\boldsymbol{x}$ belonging to class $k$. We now investigate the statistical properties of the logit gradients $\boldsymbol{\partial}_{\boldsymbol{x}}f_k$, and then see how they shape $\partial_x\mathcal{L}$.

**Step 1: Statistical properties of $\boldsymbol{\partial}_{\boldsymbol{x}}f_k$.** Let $\mathcal{P}(x,k)$ be the set of paths $\boldsymbol{p}$ from input neuron $x$ to output-logit $k$. Let $p-1$ and $p$ be two successive neurons on path $\boldsymbol{p}$, and $\tilde{\boldsymbol{p}}$ be the same path $\boldsymbol{p}$ but without its input neuron. Let $w_p$ designate the weight from $p-1$ to $p$ and $\omega_{\boldsymbol{p}}$ be the *path-product* $\omega_{\boldsymbol{p}} := \prod_{p\in\tilde{\boldsymbol{p}}} w_p$. Finally, let $\sigma_p$ (resp. $\sigma_{\boldsymbol{p}}$) be equal to 1 if the ReLU of node $p$ (resp. if path $\boldsymbol{p}$) is active for input $\boldsymbol{x}$, and 0 otherwise.

As previously noticed by Balduzzi et al. (2017) using the chain rule, we see that $\partial_x f_k$ is the sum of all $\omega_{\boldsymbol{p}}$ whose path is active, i.e. $\partial_x f_k(\boldsymbol{x}) = \sum_{\boldsymbol{p}\in\mathcal{P}(x,k)} \omega_{\boldsymbol{p}}\sigma_{\boldsymbol{p}}$. Consequently:

$$\mathbb{E}_{W,\sigma}\left[\partial_x f_k(\boldsymbol{x})^2\right] = \sum_{\boldsymbol{p}\in\mathcal{P}(x,k)} \prod_{p\in\tilde{\boldsymbol{p}}} \mathbb{E}_W\left[w_p^2\right]\mathbb{E}_\sigma\left[\sigma_p^2\right]$$

$$= |\mathcal{P}(x,k)| \prod_{p\in\tilde{\boldsymbol{p}}} \frac{2}{d_{p-1}}\frac{1}{2} = \prod_{p\in\tilde{\boldsymbol{p}}} d_p \cdot \prod_{p\in\tilde{\boldsymbol{p}}} \frac{1}{d_{p-1}} = \frac{1}{d}\,. \quad (8)$$

The first equality uses H1 to decouple the expectations over weights and ReLUs, and then applies Lemma 10 of Appendix B.3, which uses H3-H5 to kill all cross-terms and take the expectation over weights inside the product. The second equality uses H3 and the fact that the resulting product is the same for all active paths. The third equality counts the number of paths from $x$ to $k$ and we conclude by noting that all terms cancel out, except $d_{p-1}$ from the input layer which is $d$. Equation 8 shows that $|\partial_x f_k| \propto 1/\sqrt{d}$.

**Step 2: Statistical properties of $\partial_k\mathcal{L}$ and $\partial_x\mathcal{L}$.** Defining $q_k(\boldsymbol{x}) := \frac{e^{f_k(\boldsymbol{x})}}{\sum_{h=1}^{K} e^{f_h(\boldsymbol{x})}}$ (the probability of image $\boldsymbol{x}$ belonging to class $k$ according to the network), we have, by definition of the cross-entropy loss, $\mathcal{L}(\boldsymbol{x},c) := -\log q_c(\boldsymbol{x})$, where $c$ is the label of the target class. Thus:

$$\partial_k\mathcal{L}(\boldsymbol{x}) = \begin{cases} -q_k(\boldsymbol{x}) & \text{if } k \neq c \\ 1 - q_c(\boldsymbol{x}) & \text{otherwise,} \end{cases} \quad \text{and}$$

$$\partial_x\mathcal{L}(\boldsymbol{x}) = (1-q_c)\,\partial_x f_c(\boldsymbol{x}) + \sum_{k\neq c} q_k\,(-\partial_x f_k(\boldsymbol{x}))\,. \quad (9)$$

Using again Lemma 10, we see that the $\partial_x f_k(\boldsymbol{x})$ are $K$ centered and uncorrelated variables. So $\partial_x\mathcal{L}(\boldsymbol{x})$ is approximately the sum of $K$ uncorrelated variables with zero-mean, and its total variance is given by $\left((1-q_c)^2 + \sum_{k\neq c} q_k^2\right)/d$. Hence the magnitude of $\partial_x\mathcal{L}(\boldsymbol{x})$ is $1/\sqrt{d}$ for all $\boldsymbol{x}$, so the $\ell_q$-norm of the full input gradient is $d^{1/q-1/2}$. (6) concludes. $\quad \square$

*Remark* 1. Equation 9 can be rewritten as

$$\partial_x\mathcal{L}(\boldsymbol{x}) = \sum_{k=1}^{K} q_k(\boldsymbol{x})\left(\partial_x f_c(\boldsymbol{x}) - \partial_x f_k(\boldsymbol{x})\right)\,. \quad (10)$$

As the term $k=c$ disappears, the norm of the gradients $\partial_x\mathcal{L}(\boldsymbol{x})$ appears to be controlled by the total error probability. This suggests that, even without regularization, trying to decrease the ordinary

classification error is still a valid strategy against adversarial examples. It reflects the fact that when increasing the classification margin, larger gradients of the classifier's logits are needed to push images from one side of the classification boundary to the other. This is confirmed by Theorem 2.1 of Hein & Andriushchenko (2017). See also (16) in Appendix C.

## B.3 PROOF OF THEOREM 5

The proof of Theorem 5 is very similar to the one of Theorem 4, but we will need to first generalize the equalities appearing in (8). To do so, we identify the computational graph of a neural network to an abstract Directed Acyclic Graph (DAG) which we use to prove the needed algebraic equalities. We then concentrate on the statistical weight-interactions implied by assumption $(\mathcal{H})$, and finally throw these results together to prove the theorem. In all the proof, $o$ will designate one of the output-logits $f_k(\boldsymbol{x})$.

**Lemma 8.** *Let $\boldsymbol{x}$ be the vector of inputs to a given DAG, $o$ be any leaf-node of the DAG, $x$ a generic coordinate of $\boldsymbol{x}$. Let $\boldsymbol{p}$ be a path from the set of paths $\mathcal{P}(x, o)$ from $x$ to $o$, $\tilde{\boldsymbol{p}}$ the same path without node $x$, $p$ a generic node in $\tilde{\boldsymbol{p}}$, and $d_p$ be its input-degree. Then:*

$$\sum_{x \in \boldsymbol{x}} \sum_{\tilde{\boldsymbol{p}} \in \mathcal{P}(x,o)} \prod_{p \in \tilde{\boldsymbol{p}}} \frac{1}{d_p} = 1 \tag{11}$$

*Proof.* We will reason on a random walk starting at $o$ and going up the DAG by choosing any incoming node with equal probability. The DAG being finite, this walk will end up at an input-node $x$ with probability 1. Each path $\boldsymbol{p}$ is taken with probability $\prod_{p \in \tilde{\boldsymbol{p}}} \frac{1}{d_p}$. And the probability to end up at an input-node is the sum of all these probabilities, i.e. $\sum_{x \in \boldsymbol{x}} \sum_{\boldsymbol{p} \in \mathcal{P}(x,o)} \prod_{p \in \boldsymbol{p}} d_p^{-1}$, which concludes. $\square$

The sum over all inputs $x$ in (11) being 1, on average it is $1/d$ for each $x$, where $d$ is the total number of inputs (i.e. the length of $\boldsymbol{x}$). It becomes an equality under assumption $(\mathcal{S})$:

**Lemma 9.** *Under the symmetry assumption $(\mathcal{S})$, and with the previous notations, for any input $x \in \boldsymbol{x}$:*

$$\sum_{\boldsymbol{p} \in \mathcal{P}(x,o)} \prod_{p \in \tilde{\boldsymbol{p}}} \frac{1}{d_p} = \frac{1}{d} \, . \tag{12}$$

*Proof.* Let us denote $\mathcal{D}(x, o) := \{d_{\boldsymbol{p}}\}_{x \in \mathcal{P}(x,o)}$. Each path $\boldsymbol{p}$ in $\mathcal{P}(x, o)$ corresponds to exactly one element $d_{\boldsymbol{p}}$ in $\mathcal{D}(x, o)$ and vice-versa. And the elements $d_p$ of $d_{\boldsymbol{p}}$ completely determine the product $\prod_{p \in \tilde{\boldsymbol{p}}} d_p^{-1}$. By using (11) and the fact that, by $(\mathcal{S})$, the multiset $\mathcal{D}(x, o)$ is independent of $x$, we hence conclude

$$\sum_{x \in \boldsymbol{x}} \sum_{\boldsymbol{p} \in \mathcal{P}(x,o)} \prod_{p \in \tilde{\boldsymbol{p}}} \frac{1}{d_p} = \sum_{x \in \boldsymbol{x}} \sum_{d_{\boldsymbol{p}} \in \mathcal{D}(x,o)} \prod_{d_p \in d_{\boldsymbol{p}}} \frac{1}{d_p}$$

$$= d \sum_{d_{\boldsymbol{p}} \in \mathcal{D}(x,o)} \prod_{d_p \in d_{\boldsymbol{p}}} \frac{1}{d_p} = 1 \, . \qquad \square$$

Now, let us relate these considerations on graphs to gradients and use assumptions $(\mathcal{H})$. We remind that path-product $\omega_{\boldsymbol{p}}$ is the product $\prod_{p \in \tilde{\boldsymbol{p}}} w_p$.

**Lemma 10.** *Under assumptions $(\mathcal{H})$, the path-products $\omega_{\boldsymbol{p}}, \omega_{\boldsymbol{p}'}$ of two distinct paths $\boldsymbol{p}$ and $\boldsymbol{p}'$ starting from a same input node $x$, satisfy:*

$$\mathbb{E}_W \left[ \omega_{\boldsymbol{p}} \, \omega_{\boldsymbol{p}'} \right] = 0 \quad and \quad \mathbb{E}_W \left[ \omega_{\boldsymbol{p}}^2 \right] = \prod_{p \in \tilde{\boldsymbol{p}}} \mathbb{E}_W \left[ w_p^2 \right] \, .$$

*Furthermore, if there is at least one non-average-pooling weight on path $\boldsymbol{p}$, then $\mathbb{E}_W \left[ \omega_{\boldsymbol{p}} \right] = 0$.*

*Proof.* Hypothesis H4 yields

$$\mathbb{E}_W\left[\omega_{\boldsymbol{p}}^2\right] = \mathbb{E}_W\left[\prod_{p\in\tilde{\boldsymbol{p}}} w_p^2\right] = \prod_{p\in\tilde{\boldsymbol{p}}}\mathbb{E}_W\left[w_p^2\right].$$

Now, take two different paths $\boldsymbol{p}$ and $\boldsymbol{p}'$ that start at a same node $x$. Starting from $x$, consider the first node after which $\boldsymbol{p}$ and $\boldsymbol{p}'$ part and call $p$ and $p'$ the next nodes on $\boldsymbol{p}$ and $\boldsymbol{p}'$ respectively. Then the weights $w_p$ and $w_{p'}$ are two weights of a same node. Applying H4 and H5 hence gives

$$\mathbb{E}_W\left[\omega_{\boldsymbol{p}}\,\omega_{\boldsymbol{p}'}\right] = \mathbb{E}_W\left[\omega_{\boldsymbol{p}\backslash p}\,\omega_{\boldsymbol{p}'\backslash p'}\right]\mathbb{E}_W\left[w_p\,w_{p'}\right] = 0.$$

Finally, if $\boldsymbol{p}$ has at least one non-average-pooling node $p$, then successively applying H4 and H3 yields: $\mathbb{E}_W\left[\omega_{\boldsymbol{p}}\right] = \mathbb{E}_W\left[\omega_{\boldsymbol{p}\backslash p}\right]\mathbb{E}_W\left[w_p\right] = 0$. $\qquad\square$

We now have all elements to prove Theorem 5.

*Proof.* (**of Theorem 5**) For a given neuron $p$ in $\tilde{p}$, let $p-1$ designate the previous node in $\boldsymbol{p}$ of $p$. Let $\sigma_p$ (resp. $\sigma_{\boldsymbol{p}}$) be a variable equal to 0 if neuron $p$ gets killed by its ReLU (resp. path $\boldsymbol{p}$ is inactive), and 1 otherwise. Then:

$$\partial_x o = \sum_{\boldsymbol{p}\in\mathcal{P}(x,o)}\prod_{p\in\tilde{\boldsymbol{p}}}\partial_{p-1}\,p = \sum_{\boldsymbol{p}\in\mathcal{P}(x,o)}\omega_{\boldsymbol{p}}\,\sigma_{\boldsymbol{p}}$$

Consequently:

$$\begin{aligned}
\mathbb{E}_{W,\sigma}\left[(\partial_x o)^2\right] &= \sum_{\boldsymbol{p},\boldsymbol{p}'\in\mathcal{P}(x,o)}\mathbb{E}_W\left[\omega_{\boldsymbol{p}}\,\omega_{\boldsymbol{p}'}\right]\mathbb{E}_\sigma\left[\sigma_{\boldsymbol{p}}\sigma_{\boldsymbol{p}'}\right]\\
&= \sum_{\boldsymbol{p}\in\mathcal{P}(x,o)}\prod_{p\in\tilde{\boldsymbol{p}}}\mathbb{E}_W\left[\omega_p^2\right]\mathbb{E}_\sigma\left[\sigma_p^2\right] \qquad\qquad(13)\\
&= \sum_{\boldsymbol{p}\in\mathcal{P}(x,o)}\prod_{p\in\tilde{\boldsymbol{p}}}\frac{2}{d_p}\frac{1}{2} = \frac{1}{d},
\end{aligned}$$

where the firs line uses the independence between the ReLU killings and the weights (H1), the second uses Lemma 10 and the last uses Lemma 9. The gradient $\boldsymbol{\partial_x} o$ thus has coordinates whose squared expectations scale like $1/d$. Thus each coordinate scales like $1/\sqrt{d}$ and $\|\boldsymbol{\partial_x} o\|_q$ like $d^{1/2-1/q}$. Conclude on $\|\boldsymbol{\partial_x}\mathcal{L}\|_q$ and $\epsilon_p\|\boldsymbol{\partial_x}\mathcal{L}\|_q$ by using Step 2 of the proof of Theorem 4.

Finally, note that, even without the symmetry assumption ($\mathcal{S}$), using Lemma 8 shows that

$$\begin{aligned}
\mathbb{E}_W\left[\|\boldsymbol{\partial_x} o\|_2^2\right] &= \sum_{x\in\boldsymbol{x}}\mathbb{E}_W\left[(\partial_x o)^2\right]\\
&= \sum_{x\in\boldsymbol{x}}\sum_{\boldsymbol{p}\in\mathcal{P}(x,o)}\prod_{p\in\tilde{\boldsymbol{p}}}\frac{2}{d_p}\frac{1}{2} = 1.
\end{aligned}$$

Thus, with or without ($\mathcal{S}$), $\|\boldsymbol{\partial_x} o\|_2$ is independent of the input-dimension $d$. $\qquad\square$

### B.4 Proof of Theorem 7

To prove Theorem 7, we will actually prove the following more general theorem, which generalizes Theorem 5. Theorem 7 is a straightforward corollary of it.

**Theorem 11.** *Consider any feed-forward network with linear connections and ReLU activation functions that outputs logits $f_k(\boldsymbol{x})$ and satisfies assumptions ($\mathcal{H}$). Suppose that there is a fixed multiset of integers $\{a_1,\ldots,a_n\}$ such that each path from input to output traverses exactly $n$ average pooling nodes with degrees $\{a_1,\ldots,a_n\}$. Then:*

$$\|\partial_x f_k\|_2 \propto \frac{1}{\prod_{i=1}^n \sqrt{a_i}}. \qquad\qquad(14)$$

*Furthermore, if the net satisfies the symmetry assumption ($\mathcal{S}$), then: $|\partial_x f_k| \propto \frac{1}{\sqrt{d\prod_{i=1}^n a_i}}$.*

Two remarks. First, in all this proof, "weight" encompasses both the standard random weights, and the constant (deterministic) weights equal to $1/$(in-degree) of the average-poolings. Second, assumption H5 implies that the average-pooling nodes have disjoint input nodes: otherwise, there would be two non-zero deterministic weights $w, w'$ from a same neuron that would hence satisfy: $\mathbb{E}_W [w\,w'] \neq 0$.

*Proof.* As previously, let $o$ designate any fixed output-logit $f_k(\boldsymbol{x})$. For any path $\boldsymbol{p}$, let $\boldsymbol{a}$ be the set of average-pooling nodes of $\boldsymbol{p}$ and let $\boldsymbol{q}$ be the set of remaining nodes. Each path-product $\omega_{\boldsymbol{p}}$ satisfies: $\omega_{\boldsymbol{p}} = \omega_{\boldsymbol{q}}\omega_{\boldsymbol{a}}$, where $\omega_{\boldsymbol{a}}$ is a same fixed constant. For two distinct paths $\boldsymbol{p}, \boldsymbol{p}'$, Lemma 10 therefore yields: $\mathbb{E}_W \left[\omega_{\boldsymbol{p}}^2\right] = \omega_{\boldsymbol{a}}^2\,\mathbb{E}_W \left[\omega_{\boldsymbol{q}}^2\right]$ and $\mathbb{E}_W [\omega_{\boldsymbol{p}}\omega_{\boldsymbol{p}'}] = 0$. Combining this with Lemma 9 and under assumption $(\mathcal{S})$, we get similarly to (13):

$$
\begin{aligned}
\mathbb{E}_{W,\sigma} \left[(\partial_x o)^2\right] &= \sum_{\boldsymbol{p},\boldsymbol{p}'\in\mathcal{P}(x,o)} \omega_{\boldsymbol{a}}\omega_{\boldsymbol{a}'}\,\mathbb{E}_W \left[\omega_{\boldsymbol{q}}\,\omega_{\boldsymbol{q}'}\right] \mathbb{E}_\sigma \left[\sigma_{\boldsymbol{q}}\sigma_{\boldsymbol{q}'}\right] \\
&= \sum_{\boldsymbol{p}\in\mathcal{P}(x,o)} \prod_{i=1}^n \frac{1}{a_i^2} \prod_{q\in\tilde{\boldsymbol{q}}} \mathbb{E}_W \left[\omega_q^2\right] \mathbb{E}_\sigma \left[\sigma_q^2\right] \\
&= \underbrace{\prod_{i=1}^n \frac{1}{a_i}}_{\substack{\text{same value}\\\text{for all } \boldsymbol{p}}} \sum_{\boldsymbol{p}\in\mathcal{P}(x,o)} \underbrace{\prod_{i=1}^n \frac{1}{a_i} \prod_{q\in\tilde{\boldsymbol{q}}} \frac{2}{d_q} \frac{1}{2}}_{\Pi_{p\in\tilde{\boldsymbol{p}}} \frac{1}{d_p}} \\
&= \frac{1}{d} \prod_{i=1}^n \frac{1}{a_i} \, . \qquad \underbrace{\phantom{xxxxx}}_{=\frac{1}{d} \;\; \text{(Lemma 9)}}
\end{aligned}
\tag{15}
$$

Therefore, $|\partial_x o| = |\partial_x f_k| \propto 1/\sqrt{d \prod_{i=1}^n a_i}$. Again, note that, even without assumption $(\mathcal{S})$, using (15) and Lemma 8 shows that

$$
\begin{aligned}
\mathbb{E}_W \left[\|\boldsymbol{\partial_x} o\|_2^2\right] &= \sum_{x\in\boldsymbol{x}} \mathbb{E}_{W,\sigma} \left[(\partial_x o)^2\right] \\
&\overset{(15)}{=} \sum_{x\in\boldsymbol{x}} \prod_{i=1}^n \frac{1}{a_i} \sum_{\boldsymbol{p}\in\mathcal{P}(x,o)} \prod_{i=1}^n \frac{1}{a_i} \prod_{p\in\tilde{\boldsymbol{p}}} \frac{2}{d_p} \frac{1}{2} \\
&= \prod_{i=1}^n \frac{1}{a_i} \underbrace{\sum_{x\in\boldsymbol{x}} \sum_{\boldsymbol{p}\in\mathcal{P}(x,o)} \prod_{p\in\tilde{\boldsymbol{p}}} \frac{1}{d_p}}_{=1 \;(\text{Lemma 8})} = \prod_{i=1}^n \frac{1}{a_i} \, ,
\end{aligned}
$$

which proves (14). $\qquad\qquad\square$

## C  COMPARISON TO THE CROSS-LIPSCHITZ REGULARIZER

In their Theorem 2.1, Hein & Andriushchenko (2017) show that the minimal $\epsilon = \|\boldsymbol{\delta}\|_p$ perturbation to fool the classifier must be bigger than:

$$
\min_{k\neq c} \frac{f_c(\boldsymbol{x}) - f_k(\boldsymbol{x})}{\max_{y\in B(x,\epsilon)} \|\boldsymbol{\partial_x} f_c(y) - \boldsymbol{\partial_x} f_k(y)\|_q} \, .
\tag{16}
$$

They argue that the training procedure typically already tries to maximize $f_c(\boldsymbol{x}) - f_k(\boldsymbol{x})$, thus one only needs to additionally ensure that $\|\boldsymbol{\partial_x} f_c(\boldsymbol{x}) - \boldsymbol{\partial_x} f_k(\boldsymbol{x})\|_q$ is small. They then introduce what they call a Cross-Lipschitz Regularization, which corresponds to the case $p = 2$ and involves the gradient differences between *all* classes:

$$
\mathcal{R}_{\text{xLip}} := \frac{1}{K^2} \sum_{k,h=1}^K \|\boldsymbol{\partial_x} f_h(\boldsymbol{x}) - \boldsymbol{\partial_x} f_k(\boldsymbol{x})\|_2^2
\tag{17}
$$

In contrast, using (10), (the square of) our proposed regularizer $\|\boldsymbol{\partial_x}\mathcal{L}\|_q$ from (4) can be rewritten, for $p = q = 2$ as:

$$\mathcal{R}_{\|\cdot\|_2}(f) = \sum_{k,h=1}^{K} q_k(\boldsymbol{x})q_h(\boldsymbol{x}) \left( \boldsymbol{\partial_x} f_c(\boldsymbol{x}) - \boldsymbol{\partial_x} f_k(\boldsymbol{x}) \right) \cdot$$

$$\cdot \left( \boldsymbol{\partial_x} f_c(\boldsymbol{x}) - \boldsymbol{\partial_x} f_h(\boldsymbol{x}) \right) \quad (18)$$

Although both (17) and (18) consist in $K^2$ terms, corresponding to the $K^2$ cross-interaction between the $K$ classes, the big difference is that while in (17) all classes play exactly the same role, in (18) the summands all refer to the target class $c$ in at least two different ways. First, all gradient differences are always taken with respect to $\boldsymbol{\partial_x} f_c$. Second, each summand is weighted by the probabilities $q_k(\boldsymbol{x})$ and $q_h(\boldsymbol{x})$ of the two involved classes, meaning that only the classes with a non-negligible probability get their gradient regularized. This reflects the idea that only points near the margin need a gradient regularization, which incidentally will make the margin sharper.

## D    PERCEPTION THRESHOLD

To keep the average pixel-wise variation constant across dimensions $d$, we saw in (3) that the threshold $\epsilon_p$ of an $\ell_p$-attack should scale like $d^{1/p}$. We will now see another justification for this scaling. Contrary to the rest of this work, where we use a fixed $\epsilon_p$ for all images $\boldsymbol{x}$, here we will let $\epsilon_p$ depend on the $\ell_2$-norm of $\boldsymbol{x}$. If, as usual, the dataset is normalized such that the pixels have on average variance 1, both approaches are almost equivalent.

Suppose that given an $\ell_p$-attack norm, we want to choose $\epsilon_p$ such that the signal-to-noise ratio (SNR) $\|\boldsymbol{x}\|_2 / \|\boldsymbol{\delta}\|_2$ of a perturbation $\boldsymbol{\delta}$ with $\ell_p$-norm $\leq \epsilon_p$ is never greater than a given SNR threshold $1/\epsilon$. For $p = 2$ this imposes $\epsilon_2 = \epsilon \|\boldsymbol{x}\|_2$. More generally, studying the inclusion of $\ell_p$-balls in $\ell_2$-balls yields

$$\epsilon_p = \epsilon \|\boldsymbol{x}\|_2 \, d^{1/p-1/2} \, . \quad (19)$$

Note that this gives again $\epsilon_p = \epsilon_\infty d^{1/p}$. This explains how to adjust the threshold $\epsilon$ with varying $\ell_p$-attack norm.

Now, let us see how to adjust the threshold of a given $\ell_p$-norm when the dimension $d$ varies. Suppose that $\boldsymbol{x}$ is a natural image and that decreasing its dimension means either decreasing its resolution or cropping it. Because the statistics of natural images are approximately resolution and scale invariant (Huang, 2000), in either case the average squared value of the image pixels remains unchanged, which implies that $\|\boldsymbol{x}\|_2$ scales like $\sqrt{d}$. Pasting this back into (19), we again get:

$$\epsilon_p = \epsilon_\infty \, d^{1/p} \, .$$

In particular, $\epsilon_\infty \propto \epsilon$ is a dimension-free number, exactly like in (3) of the main part.

Now, why did we choose the SNR as our invariant reference quantity and not anything else? One reason is that it corresponds to a physical power ratio between the image and the perturbation, which we think the human eye is sensible to. Of course, the eye's sensitivity also depends on the spectral frequency of the signals involved, but we are only interested in orders of magnitude here.

Another point: any image $\boldsymbol{x}$ yields an adversarial perturbation $\boldsymbol{\delta_x}$, where by constraint $\|\boldsymbol{x}\|_2 / \|\boldsymbol{\delta_x}\| \leq 1/\epsilon$. For $\ell_2$-attacks, this inequality is actually an equality. But what about other $\ell_p$-attacks: (on average over $\boldsymbol{x}$,) how far is the signal-to-noise ratio from its imposed upper bound $1/\epsilon$? For $p \notin \{1, 2, \infty\}$, the answer unfortunately depends on the pixel-statistics of the images. But when $p$ is 1 or $\infty$, then the situation is locally the same as for $p = 2$. Specifically:

**Lemma 12.** *Let $\boldsymbol{x}$ be a given input and $\epsilon > 0$. Let $\epsilon_p$ be the greatest threshold such that for any $\boldsymbol{\delta}$ with $\|\boldsymbol{\delta}\|_p \leq \epsilon_p$, the SNR $\|\boldsymbol{x}\|_2 / \|\boldsymbol{\delta}\|_2$ is $\leq 1/\epsilon$. Then $\epsilon_p = \epsilon \|\boldsymbol{x}\|_2 \, d^{1/p-1/2}$.*

*Moreover, for $p \in \{1, 2, \infty\}$, if $\boldsymbol{\delta_x}$ is the $\epsilon_p$-sized $\ell_p$-attack that locally maximizes the loss-increase i.e. $\boldsymbol{\delta_x} = \arg\max_{\|\boldsymbol{\delta}\|_p \leq \epsilon_p} |\boldsymbol{\partial_x}\mathcal{L} \cdot \boldsymbol{\delta}|$, then:*

$$\text{SNR}(\boldsymbol{x}) := \frac{\|\boldsymbol{x}\|_2}{\|\boldsymbol{\delta_x}\|_2} = \frac{1}{\epsilon} \quad and \quad \mathbb{E}_{\boldsymbol{x}}\left[\text{SNR}(\boldsymbol{x})\right] = \frac{1}{\epsilon} \, .$$

*Proof.* The first paragraph follows from the fact that the greatest $\ell_p$-ball included in an $\ell_2$-ball of radius $\epsilon \|\boldsymbol{x}\|_2$ has radius $\epsilon \|\boldsymbol{x}\|_2 \, d^{1/p-1/2}$.

The second paragraph is clear for $p = 2$. For $p = \infty$, it follows from the fact that $\boldsymbol{\delta_x} = \epsilon_\infty \operatorname{sign} \boldsymbol{\partial_x}\mathcal{L}$ which satisfies: $\|\boldsymbol{\delta_x}\|_2 = \epsilon_\infty \sqrt{d} = \epsilon \|\boldsymbol{x}\|_2$. For $p = 1$, it is because $\boldsymbol{\delta_x} = \epsilon_1 \max_{i=1..d} |(\boldsymbol{\partial_x}\mathcal{L})_i|$, which satisfies: $\|\boldsymbol{\delta_x}\|_2 = \epsilon_2/\sqrt{d} = \epsilon \|\boldsymbol{x}\|_2$. $\qquad\square$

Intuitively, this means that for $p \in \{1, 2, \infty\}$, the SNR of $\epsilon_p$-sized $\ell_p$-attacks on any input $\boldsymbol{x}$ will be exactly equal to its fixed upper limit $1/\epsilon$. And in particular, the mean SNR over samples $\boldsymbol{x}$ is the same $(1/\epsilon)$ in all three cases.

# E  VULNERABILITY-DIMENSION DEPENDENCE USING DOWNSIZED IMAGENET IMAGES

We also ran a similar experiment as in Section 4.2, but instead of using upsampled CIFAR-10 images, we created a 12-class dataset of approximately $80,000$ $3 \times 256 \times 256$-sized RGB-images by merging similar ImageNet-classes, resizing the smallest image-edge to 256 pixels and center-cropping the result. We then downsized the images to 32, 64, 128 and 256 pixels per edge, and trained, not 1, but *10 CNNs per image-size*. We then computed their adversarial vulnerability and average $\|\boldsymbol{\partial_x}\mathcal{L}\|_1$. This gave us 2 values per trained net, i.e. 2 x 10 values per image-size, which are shown in 6. The lines follow their medians, the errorbars show their $10^{\text{th}}$ and $90^{\text{th}}$ quantiles. The conclusions are identical to Section 4.2: after usual training, the vulnerability and gradient-norms still increase like

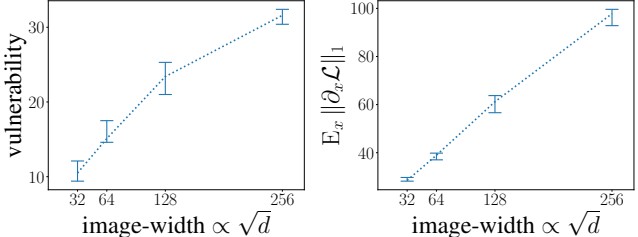

Figure 6: Similar experiment than Fig 3, but with downsampled ImageNet images, rather than upsampled CIFAR-10 images. Same conclusion: after usual training, vulnerability and average $\|\boldsymbol{\partial_x}\mathcal{L}\|_1$ grow like $\sqrt{d}$.

$\sqrt{d}$. Note that, as the gradients get much larger at higher dimensions, the first order approximation in (2) becomes less and less valid, which explains the little inflection of the adversarial vulnerability curve. For smaller $\epsilon$-thresholds, we verified that the inflection disappears.

# F  FIGURES WITH AN $\ell_2$ PERTURBATION-THRESHOLD AND DEEP-FOOL ATTACKS

Here we plot the same curves as in the main part, but using an $\ell_2$-attack threshold of size $\epsilon_2 = 0.005\sqrt{d}$ instead of the $\ell_\infty$-threshold and deep-fool attacks (Moosavi-Dezfooli et al., 2016) instead of iterative $\ell_\infty$-ones in Figs. 8 and 9. Note that contrary to $\ell_\infty$-thresholds, $\ell_2$-thresholds must be rescaled by $\sqrt{d}$ to stay consistent across dimensions (see Eq.3 and Appendix D). All curves look essentially the same as their counterparts in the main text.

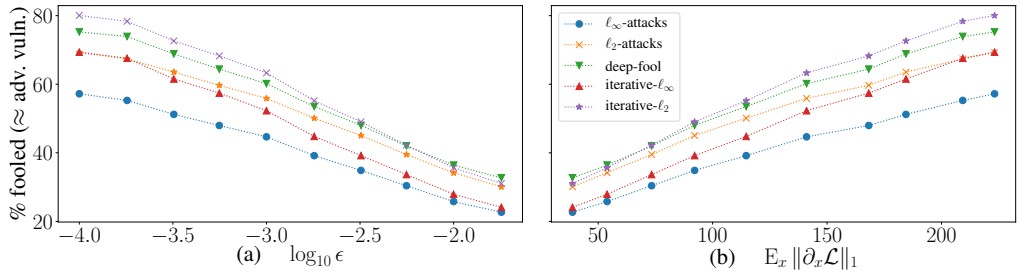

Figure 7: Same as Figure 1 but using an $\ell_2$ threshold instead of a $\ell_\infty$ one. Now the $\ell_2$-based methods (deep-fool, and single-step and iterative $\ell_2$-attacks) seem more effective than the $\ell_\infty$ ones.

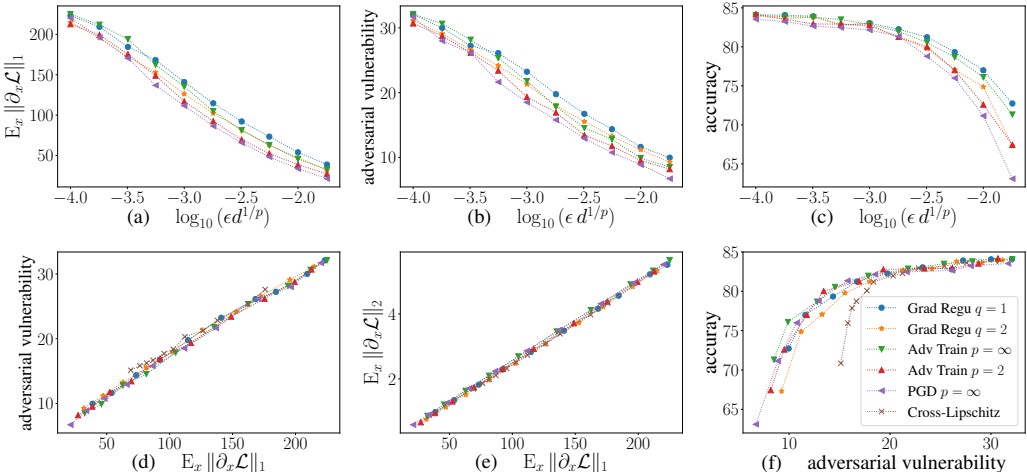

Figure 8: Same as Figure 2, but with an $\ell_2$- perturbation-threshold (instead of $\ell_\infty$) and deep-fool attacks (Moosavi-Dezfooli et al., 2016) instead of iterative $\ell_\infty$ ones. All curves look essentially the same than in Fig. 2.

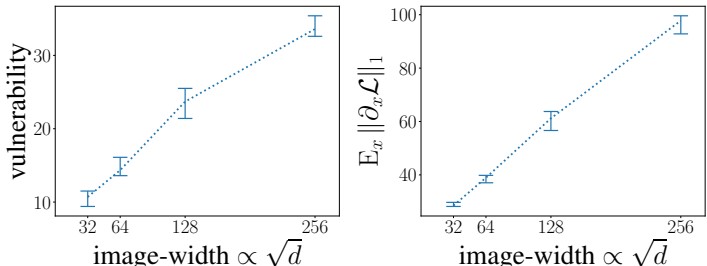

Figure 9: Same as Figure 6 but with an $\ell_2$ perturbation-threshold (instead of an $\ell_\infty$ one) and using deep-fool (instead of iterative-$\ell_\infty$) attacks to approximate adversarial vulnerability.

## G  A VARIANT OF ADVERSARIALLY-AUGMENTED TRAINING

In usual adversarially-augmented training, the adversarial image $x + \delta$ is generated on the fly, but is nevertheless treated as a fixed input of the neural net, which means that the gradient does not get backpropagated through $\delta$. This need not be. As $\delta$ is itself a function of $x$, the gradients could actually also be backpropagated through $\delta$. As it was only a one-line change of our code, we used this opportunity to test this variant of adversarial training (FGSM-variant in Figure 2) and thank Martín Arjovsky for suggesting it. But except for an increased computation time, we found no significant difference compared to usual augmented training.

