# OpenReview forum: "Adversarial Vulnerability of Neural Networks Increases with Input Dimension"
_ICLR.cc/2019/Conference_

### Official Review · AnonReviewer2 · 2018-10-30
**interesting work but with limited applicability and significance demonstrated**

**Rating:** 5
**Confidence:** 5

**Review:**

This paper analyzes the relationship between "adversarial vulnerability" with input dimensionality of neural network. The paper proves that, under certain assumptions, as the input dimensionality increases, neural networks exhibit increasingly large gradients thus are more adversarially vulnerable. Experiments were done on neural networks trained by penalizing input gradients and FGSM-adversarial training. Similar trends on vulnerability vs dimensionality are found.

The paper is clearly written and easy to follow. I appreciate that the authors also clearly stated the limitation of the theoretical analysis.

The theoretical analyses on vulnerability and dimensionality is novel and provide some insights. But it is unlikely such analysis is significant There are a few reasons:
- This analysis only seems to work for "well-behaved" models. For models with gradient masking, obfuscated gradients or even non-differentiable models, it is not clear that how this will apply. (and I appreciate that the authors also acknowledge this in the paper.) It is unclear how this specific gradient based analysis can help the understanding of the adversarial perturbation phenomena. After all, the first order Taylor expansion argument on top of randomly initialized weights is oversimplifying the complicated problem.
- One very important special case of the point above: the analysis probably cannot cover the  adversarially PGD trained models [MMS+17] and the certifiably robust ones. Such models may have small gradients inside the box constraint, but can have large gradients between different classes.


On the empirical results, the authors made a few interesting observations, for example the close correspondence between "Adv Train" and "Grad Regu" models.
My concern is that the experiments were done on a narrow range of models, which only have "weak" adversarial training / defenses.
Adversarial robustness is hard to achieve. What matters the most is "why the strongest model is still not robust?" not "why some weak models are not robust?"
It is especially worrisome to me that the paper does not cover the adversarially-augmented training based iterative attacks, e.g. PGD TRAINED models [MMS+17] which is the SOTA on MNIST/CIFAR10 L_\infty robustness benchmark.
Without comprehensive analyses on SOTA robust models, it is hard to justify the validity of the theoretical analysis in this paper, and the conclusions made by the paper.
For example, re: the last sentence in the conclusion: "They hence suggest to tackle adversarial vulnerability by designing new architectures (or new architectural building blocks) rather than by new regularization techniques." The reasoning is not obvious to me given the current evidence shown in the paper.

[MMS+17] Madry A, Makelov A, Schmidt L, Tsipras D, Vladu A. Towards deep learning models resistant to adversarial attacks. arXiv preprint arXiv:1706.06083

---

> ### Author Response · Authors · 2018-11-19
> **Specific answers to your comments & concerns**
>
> We thank you for your time, review, comments and concerns, which we hope to address in full.
>
> - Concerning: "This analysis only seems to work for 'well-behaved' models. For models ... apply"
>
> Indeed, we only analyse differentiable models. First, note that our results already cover many usual networks (not just a small subset). Second, we think that understanding such well-behaved models is a first step towards understanding non-differentiable ones. (For example, some non-differentiable functions can be considered differentiable at a rougher scale. But this opens a whole new research direction, while the text is long enough...)
>
> - Concerning: "the first order Taylor expansion argument on top of randomly initialized weights is oversimplifying the complicated problem."
>
> Not all adversarial vulnerability might be first-order, but first-order vulnerability *is* an aspect of vulnerability (not an oversimplification of a problem). If there is first-order vulnerability, then there is vulnerability. Moreover, our results actually suggest that first-order vulnerability and its relation to gradients explains an *essential* part of vulnerability (see Fig 2d, and paragraph "Validity of first order expansion").
>
> - Concerning: "the analysis probably cannot cover the  adversarially PGD trained models [MMS+17] and the certifiably robust ones" & "It is especially worrisome to me that the paper does not cover the adversarially-augmented training based iterative attacks, e.g. PGD TRAINED models"
>
> We added experiments with PGD training on CIFAR-10 (see Fig 2 & 6). Our conclusions stay unchanged. The new experiments support our claim that first-order vulnerability plays an essential role.
>
> - Concerning: "What matters the most is 'why the strongest model is still not robust?' not 'why some weak models are not robust?'"
>
> We think that understanding the vulnerability of "weak" models (i.e. at initialization or with usual training) may help understanding the vulnerability of SOTA-robustly trained nets. See our post on "why prior vulnerability matters".
>
> - Concerning our last sentence:
>
> We can reformulate it to:
> "Nevertheless, they show that at least this type of first-order vulnerability is present, common, and firmly rooted *in the priors* of our current network architectures. In future, we may hence want to complement our robust regularisation techniques by new architectures (or architectural building blocks) with less vulnerable priors."
> (Anything in that direction would do. We are open to propositions.)

---

### Official Review · AnonReviewer1 · 2018-11-02
**A solid contribution to the study of adversarial examples.**

**Rating:** 9
**Confidence:** 4

**Review:**

The authors provide a compelling theoretical explanation for a large class of adversarial examples.  While this explanation (rooted in the norm of gradients of neural networks being the culprit for the existence of adversarial examples) is not new, they unify several old perspectives, and convincingly argue for genuinely new scaling relationships (i.e. \sqrt(d) versus linear in d scaling of sensitivity to adversarial perturbations versus input size). They prove a number of theorems relating these scaling relationships to a broad swathe of relevant model architectures, and provide thorough empirical evidence of their work.

I can honestly find very little to complain about in this work--the prose is clear, and the proofs are correct as far as I can tell (though I found Figure 4 in the appendix (left panel) to not be hugely compelling.  More data here would be great!)

As much of the analysis hinges on the particularities of the weight distribution at initialization, could the authors comment on possible defenses to adversarial attack by altering this weight distribution? (By, for example, imposing that the average value must grow like 1/d)?

---

> ### Author Response · Authors · 2018-11-19
> **Specific answers to your few questions**
>
> We thank you for your review and your very positive evaluation.
>
> Concerning Fig 4 in Appendix A:
> Appendix A is preliminary work, whose goal is essentially to illustrate how our insights on the prior-vulnerability of neural networks can help us design more robust networks; in this case, by preferring average-poolings over other pooling-operations. But we agree that this section only contains preliminary results, which is why it is in appendix, not main text.
>
> Concerning: "could the authors comment on possible defences to adversarial attack by altering this weight distribution? (By, for example, imposing that the average value must grow like 1/d)?"
> Please refer to point 3/ of our overall reply, which explains what problems arise if we just change the overall weight-size at init.
>
> We hope that this addresses your small concerns/questions and thank you, once again, for your evaluation.

---

### Official Review · AnonReviewer3 · 2018-11-05
**Considered question seems poorly motivated, significance of analysis and conclusions yet to be demonstrated**

**Rating:** 4
**Confidence:** 5

**Review:**

This paper argues that adversarial vulnerability of neural networks increases with input dimension. Theoretical and empirical evidence are given which connect the l_p norm of the gradient of the training objective with the existence of small-worst case l_q perturbations. This connection is made by assuming that the learned function is well approximated by a linear function local to the sampled input x. By making assumptions on the initialization scheme for some simple architectures, the authors show that the l_p norm of the gradient for randomly initialized network will be large, and provide empirical evidence that these assumptions hold after training. These assumptions imply bounds on the typical magnitude of the gradient of the loss with respect to a single input coordinate, this then implies that the overall gradient norm will depend on the input dimension.

I found this paper well written. The mathematical assumptions are presented in a clear, easy to understand manner. Also high level intuition is given around their main theorems which help the reader understand the main ideas. However, I have a  number of concerns about this work.

The first is, I do not buy the motivation for studying the "phenomenon" of small worst-case l_p perturbations. I realize this statement applies to a large body of literature, but since the publication of  [1] we are still lacking concrete motivating scenarios for the l_p action space. I would encourage the authors instead to ask the closely related but more general question of how we can improve model generalization outside the natural distribution of images, such as generalization in the presence of commonly occurring image corruptions [2]. It's possible that the analysis in this work could better our understanding model generalization in the presence of different image corruptions, indeed by making similar linearity assumptions as considered in this work, test error in additive Gaussian noise can be linked with distance to the decision boundary [3,4]. However, this particular question was not explored in this work.

Second, the work is one of many to relate the norm of the gradient with adversarial robustness (for example, this has been proposed as a defense mechanism in [5,6]). I also suspect that the main theorem relating gradient norm to initialization should easily follow for more general settings using the mean field theory developed by [7,8] (this would be particularly useful for removing assumption H1, which assumes the ReLU activation is a random variable independent of the weights). Overall, I don't see how gradient norms explain why statistical classifiers make mistakes, particularly for more realistic attacker action spaces [9]. Even for "small" l_p adversarial examples there seem to be limitations as to how much gradient norms can explain the phenomenon --- for example even max margin classifiers such as SVM's have "adversarial examples". Furthermore, adversarial training has been shown to reach a point where the model is "robust" locally to training points but this robustness does not generalize to the points in the test set [10]. In fact, for the synthetic data distributions considered in [10], it's proven that no learning algorithm can achieve robustness given insufficient training data.

Finally, the main conclusion of this work "adversarial vulnerability of neural networks increases with input dimension" is an overly general statement which needs a much more nuanced view. While experiments shown in [11] support this conclusion for naturally trained networks, it is shown that when adversarial training is applied the model is more robust when the input dimension is higher (see Figure 4 a. and b.). Perhaps the assumptions for Theorem 4 are violated for these adversarially trained models.

1. https://arxiv.org/abs/1807.06732
2. https://arxiv.org/abs/1807.01697
3. https://arxiv.org/abs/1608.08967
4. https://openreview.net/forum?id=S1xoy3CcYX&noteId=BklKxJBF57.
5. https://arxiv.org/abs/1704.08847
6. https://arxiv.org/abs/1608.07690
7. https://arxiv.org/abs/1611.01232
8. https://arxiv.org/abs/1806.05393
9. https://arxiv.org/abs/1712.09665
10. https://arxiv.org/abs/1804.11285
11. https://arxiv.org/pdf/1809.02104.pdf

---

> ### Comment · AnonReviewer4 · 2018-11-17
> **The criticism of studying L_p perturbations is unjustified.**
>
> I understand and agree with the argument that robustness to small L_p perturbations is by no means a meaningful security guarantee. Whether this area of research is relevant to ML security is a topic of active debate that is beyond the scope of this review. However, understanding the robustness of models to small L_p perturbations is important for a variety of reasons. I will try and outline some of them below.
>
> Clearly, the vulnerability of modern ML classifiers to imperceptible perturbations is concerning as it implies that:
> a. They are not worst-case invariant to small, simple perturbations.
> b. They depend on their input in ways that we would not expect/want them to.
> (c. There could be security vulnerabilities that would evade human supervision.)
>
> So the natural question to ask is: "Are L_p adversarial examples inevitable? If so, for which models/datasets?"
> We  still do not know the answers to these questions.
> -- If this vulnerability is indeed inherent, then what are meaningful notions of worst-case invariances that our classifiers should satisfy? Even then, does our partial progress towards L_p robustness help us develop tools for different problems?
> -- If this is simply a limitation of our current models and methods, then we will eventually be able to create L_p-robust classifiers for large datasets. This would help us better understand how to enforce invariances to our model. We could then start working towards broader families of perturbations that we want to be robust to. But before any of this happens we need to understand if we can at least solve the (conceptually) very simple problem of "Can we be robust to small Lp norms?". Moreover, if we construct models that are Lp robust, are these models more useful for standard tasks in some way?
>
> We could have a very lengthy discussion about the topic, but this is clearly not the right place for that. Arguably, this topic of research is of interest to a sizeable part of the ICLR community. I would thus encourage the reviewer to focus on the technical content of the paper and let the AC decide on whether L_p robustness is a topic of interest for ICLR.

---

> > ### Comment · AnonReviewer3 · 2018-11-18
> > **Separate thread to discuss motivation**
> >
> > I'm creating a separate thread to discuss motivation, I'm willing to evaluate this paper on the technical aspects but motivation is important and the authors of this work could have asked a very related and better motivated question and written essentially the same paper.
> >
> > Why focus on small lp perturbations instead of studying the broader phenomenon that convolutional neural networks significantly underperform humans when classifying distorted images: https://arxiv.org/abs/1705.02498, https://openreview.net/forum?id=HJz6tiCqYm . The fact that models are not perfect in the presence of random image corruptions is concerning as it implies that:
> >
> > a. They are not even average case invariant to moderate simple perturbations.
> > b. They depend on their input in ways that we would not expect/want them to.
> > c. There are security vulnerabilities that will affect classifiers in many realistic deployed settings, say a street sign classifier misclassifying an input because it is a rainy day.
> > d. We will never be robust in worst-case settings until we are first robust in the average case.
> >
> > Moreover it has been known for some time that the sensitivity of models to small l2 perturbations is the same phenomenon as the sensitivity of models in the presence of large average case corruptions, see this 2016 NIPS paper: https://papers.nips.cc/paper/6331-robustness-of-classifiers-from-adversarial-to-random-noise. So your paper is essentially studying the same phenomenon that I am recommending you study, you have just chosen an odd, difficult to motivate, and difficult to evaluate metric for measuring the robustness of image classifiers. If you find small worst case perturbations surprising and interesting, but moderate average case perturbations not surprising or interesting, then I can help clarify the connection.
> >
> > Overall, my position is not so much that lp adversarial examples are uninteresting, it's just every adversarial example paper that only focuses on lp perturbations rather than studying model generalization in non-iid settings is artificially limiting the impact and scope of their work.

---

> > ### Comment · AnonReviewer3 · 2018-11-18
> > **My score holds regardless of the motivation question, here are my technical questions/concerns.**
> >
> > The experiments for adversarially trained models in [1] directly contradict the title of this paper. The models trained on the higher dimensional space are more robust (see Figure 4 b.). Can the authors comment on this? My understanding is that for some settings of the weights you can show a bound such as is discussed in the paper, but there are other settings (perhaps even initializations) of the weights for which the conclusion will not hold. To me this suggests that a more appropriate title would be “improper initialization of neural networks can cause sensitivity to small perturbations". However, fixing the initialization seems unlikely to buy us much more than what adversarial training achieves, and the experiments in [1] suggest to me the conclusion of this work is limited in scope. Indeed, even adversarially trained models are still sensitive to "small" perturbations, only the epsilon at which they are sensitive to increases slightly.
> >
> > As you mention, many prior works have explored gradient penalties as a way to increase robustness, and some have perhaps helped a little bit as an adversarial defense, but we hit a limit as we increase the epsilon considered for the perturbations, and its not clear whether or not this can improve upon adversarially trained models. Because of this, it’s not clear to me what actionable insights we can conclude from this work, and how this can be used to improve upon the current SOTA. In fact, it was found that adversarial training eventually gives robustness to the training set, but this robustness does not generalize to the test set [2]. For these models, the gradients are well behaved local to the training points (and thus any gradient based loss function will be minimized for the training points) but the gradients aren’t well behaved for new iid samples from the data distribution.
> >
> > Furthermore, in [2] it was shown that there is no learning algorithm can become robust to small perturbations, unless that model is trained on significantly more data. So at least for the synthetic data distribution they consider there is no  data independent initialization scheme that can achieve robustness.
> >
> > Minor nit: You might be able to remove the unrealistic assumption H1 from Theorem 4 by considering the theory from [3].
> >
> > 1. https://arxiv.org/abs/1809.02104
> > 2. https://arxiv.org/abs/1804.11285
> > 3. https://arxiv.org/abs/1611.01232

---

> > > ### Author Response · Authors · 2018-11-19
> > > **Specific answers to your comments & concerns**
> > >
> > > We thank you for your expanded reviews, comments, questions and references, which we hope to address in full.
> > >
> > > -Concerning: "The experiments for adversarially trained models in [1] directly contradict the title of this paper"
> > >
> > > See point 2/ of our overall reply.
> > >
> > > - Concerning: "for some settings of the weights you can show a bound such as is discussed in the paper, but there are other settings (perhaps even initializations) of the weights for which the conclusion will not hold."
> > >
> > > The current initialization-methods are used to avoid exploding/vanishing activations at init. Any other initialization would need to solve that issue. See point 3/ of our overall reply.
> > >
> > > - Concerning: "fixing the initialization seems unlikely to buy us much more than what adversarial training achieves, and the experiments in [1] suggest to me the conclusion of this work is limited in scope"
> > >
> > > Please refer to our overall thread on "why prior vulnerability matters" and how it might help understanding and harnessing the vulnerability of (robustly) trained networks.
> > >
> > > - Concerning: "we hit a limit as we increase the epsilon considered for the perturbations"
> > >
> > > Even for small epsilons, our networks are surprisingly vulnerable. If we don't understand the small epsilon vulnerability, then we won't understand big epsilons.
> > >
> > > - Concerning: "it’s not clear to me what actionable insights we can conclude from this work, and how this can be used to improve upon the current SOTA."
> > >
> > > Again, please see our thread on "why prior vulnerability matters"  and how it may help understanding and harnessing the vulnerability of (robustly) trained networks.
> > >
> > > - Concerning: "it was found that adversarial training eventually gives robustness to the training set, but this robustness does not generalize to the test set [2]... the data distribution."
> > >
> > > See 4/ in our overall reply.
> > >
> > > - Concerning: "in [2] it was shown that there is no learning algorithm [that] can become robust to small perturbations, unless that model is trained on significantly more data."
> > >
> > > Please refer to our thread on "why prior vulnerability matters". As we explain there, to get better generalisation you can either increase your amount of training data, or decrease the complexity of your model, i.e. choose better (non-vulnerable!) priors.
> > >
> > > - Concerning assumption H1 and reference [3]:
> > >
> > > the mean field approach of [3] relies on very strong independence approximations, namely, neglecting individual effects and replacing them with overall averaged effects with similar statistics. This amounts to disregarding most correlations. We do believe a mean-field treatment of our approach is possible, but in the end, the mean field approximations are much stronger than our assumption H1, though similar in spirit.
> > >
> > > [1] Are adversarial examples inevitable?, 2018
> > > [3] Deep Information Propagation, Schoenholz et al., 2017

---

> > > > ### Comment · AnonReviewer3 · 2018-11-27
> > > > **Thanks for the paper update, I still have concerns regarding the discussion of Schmidt et. al.**
> > > >
> > > > In order to improve upon the bound of Schmidt et. al. one needs a data dependent prior. Simply enforcing additional smoothness by gradient penalization is something that would apply to any dataset, so I don't see how this can explain the remaining sensitivity to test points for the adversarially trained model. Simply observing that these test points have bad gradients adds no information, of course test points which remain sensitive should be expected to have larger gradient, you're saying the model isn't robust because it isn't robust. We can't force perfect smoothness because the constant function will not fit the data, and there are settings where we may be approaching the limit of what smoothness assumptions can buy. For MNIST there are points of different classes within l2 distance of 3 of each other, and adversarially trained models have been shown to have robustness comparable to that distance.
> > > >
> > > > More experiments are needed to show that data independent gradient regularization can improve upon adversarial training.

---

### Official Review · AnonReviewer4 · 2018-11-16
**An interesting approach**

**Rating:** 6
**Confidence:** 4

**Review:**

The paper studies how the vulnerability of a neural network model depends on its input dimension. The authors prove that for an *untrained* model, randomly initialized with Xavier initialization, the gradient of the loss wrt the input is essentially independent of the architecture and task. This implies that the major factor affecting the norm of that gradient is the input dimension. They then support their argument by experiments measuring the relation between adversarial vulnerability and gradient norm using various *trained* models (including adversarially regularized ones).

I find the main theoretical result interesting. While this is a known fact for the simple case of linear classifiers, extending it to arbitrarily deep networks is a valuable contribution. The proof crucially relies on properties of the specific initialization scheme to show that the gradient does not change too much during backproparagation through the layers. The most significant limitation of the result (which the authors kindly acknowledge) is that this result only holds at initialization. Hence it cannot distinguish between different training methods or between how different architectures evolve during training. Since the situation in adversarial robustness is much more nuanced, I am skeptical about the significance of such statements.

On the experimental side, the finding that gradient regularization improves adversarial robustness to small epsilon values has been made multiple times in the past (as the authors cite in the related work section). It is worth noting that the epsilon considered is 0.005 in L_inf (1.275/255) which is pretty small. This value corresponds to the "small-epsilon regime" where the behavior of the model is fairly linear around the original inputs and thus defenses such as FGSM-training and gradient regularization are effective.

The authors also perform an interesting experiment where they train models on downsampled ImageNet datasets and find that indeed larger input dimension leads to more vulnerable models.

While I find the results interesting, I do not see clear implications. The fact that the vulnerability of a classifier depends on the L1 norm of the input gradient is already known for any locally linear classifier (i.e. deep models too), and it is fairly clear that the L1 norm will have a dimension dependence. The fact that it does not depend on architecture or task at initialization is interesting but of limited significance in my opinion. Given that the experimental results are also not particularly novel, I recommend rejection.

[UPDATE]: Given the overall discussion and paper updates, I consider the current version of the paper (marginally) crossing the ICLR bar. I update my score from a 5 to a 6.

Minor comments to the authors:
-- I think || x ||_* is more clear than |||x||| for the dual norm.
-- Consider using lambda for the regularization, epsilon is confusing since it is overloaded.

---

> ### Author Response · Authors · 2018-11-19
> **Some specific answers to your comments**
>
> We thank you for your careful review, and for pointing out that many people do indeed care about small worst-case l_p-perturbations, and why.
>
> - Concerning: "I am skeptical about the significance of such statements [at initialization]." & "While I find the results interesting, I do not see clear implications."
>
> Please refer to our overall thread on, why understanding the vulnerability of priors helps understanding post (robust) training vulnerability. See also point 4/ in our overall reply.
>
> - Concerning: "While this is a known fact for the simple case of linear classifiers..."
>
> Even with only linear classifiers, previous published work has predicted a linear increase of vulnerability with input-dimension rather than sqrt(d), because they did not take the dimension-dependance of the weights into account.
>
> - Concerning: "it is fairly clear that the L1 norm will have a dimension dependence"
>
> Maybe, but it is all about getting the numbers right. Our predictions correspond to the *exact* increase-rate measured in practice.
>
> - Concerning: "The fact that it does not depend on architecture or task at initialization is interesting but of limited significance in my opinion."
>
> This independence on architecture at initialization shows that, if we want to get non-vulnerable priors, we need to re-think our initialization scheme and/or introduce a new architectural building block. As to why we would want non-vulnerable priors, again, please see our overall thread on the subject.
>
> Once again, we thank you for your review and hope that our answers may help you to re-evaluate the significance of our results.

---

> > ### Comment · AnonReviewer4 · 2018-11-21
> > **Reviewer response**
> >
> > I appreciate the author's response to my review and their overall reply.
> >
> > I find the fact that the same linear relation holds for adversarially robust models interesting. The authors raise an additional point in their reply, namely that robust models do not generalize to the test set because the gradients are not small for test examples (despite being small for training examples). While I find this possibility intriguing, I don't see it being supported experimentally yet. If this was true, we would be able to differentiate between test examples that are vulnerable and test examples that are not based on their gradient norm, right?
> >
> > The authors argue that their results open a new direction towards training robust models by using different initialization priors. I agree that this is an interesting conjecture. However, I do not find this argument supported by the current findings of the paper. As such, I cannot consider this a contribution of this paper. It is the author's responsibility to provide arguments supporting the potential of this research direction.
> >
> > Given the overall discussion and paper edits, I consider the current version of the paper (marginally) crossing the ICLR bar and have updated my score to reflect that. Nevertheless, I would encourage the authors to update the discussion in their paper since many of the points raised here are more nuanced than those made in the paper.

---

> > > ### Author Response · Authors · 2018-11-21
> > > **Thank you for your comments & new preliminary experiments to strengthen our claim**
> > >
> > > Thank you for your quick acknowledgment of our changes and your new comments.
> > >
> > > Concerning:
> > > "The authors raise an additional point in their reply, namely that robust models do not generalize to the test set because the gradients are not small for test examples (despite being small for training examples). While I find this possibility intriguing, I don't see it being supported experimentally yet."
> > >
> > > Many thanks for this question. During the last days, we have run some experiments to assay exactly this point.
> > >
> > > We essentially re-ran the CIFAR-10 experiments (Figs 1&2) with pgd training and tracked the evolution (over training epochs) of the average gradient-norm on training and test sets respectively. The (preliminary) results are unmistakable: there is a clear discrepancy between the gradient-norms on train and test sets. On the training set, they first increase a bit, but then neatly decrease during training; on the test set however, they constantly increase.
> > > We also ran these experiments with up-sampled CIFAR-10 images (by copying pixels). The higher the resolution, the bigger the discrepancy between training and test set norms.
> > > These observations perfectly fit those of [2] (SOTA robust training reduces adversarial vulnerability on test but not on training set) and the idea that a significant amount of adversarial vulnerability can be explained by large gradients.
> > >
> > > We will add those experiments as soon as they are finished and double-checked (i.e. probably only in the final version, after the decision, since we have no material time to do this well in the coming days). We appreciate the discussion and believe it will substantially strengthen the paper. The discussion (both in content and level) illustrates that this topic is of major interest to the community, and we are confident that the final paper will satisfy your (and our) standards of mature science.
> > >
> > > [2] Adversarially Robust Generalization Requires More Data, Schmidt et al., 2018

---

> > > > ### Comment · AnonReviewer4 · 2018-12-03
> > > > **Reviewer response**
> > > >
> > > > I appreciate the authors engaging in the discussion and performing additional experiments. I do think that the additional experiments strengthen the paper. In fact I think that plotting gradient norms over training is _crucial_ to support the argument that the situation at initialization is predictive of the situation at convergence.
> > > >
> > > > Moreover, I think that during the discussion here, the authors raised interesting points that are more nuanced than the discussion presented in the paper. I would encourage them to continue working on the paper incorporating the new experiments and discussion (this is independent of the final ICLR decision).
> > > >
> > > > Finally, I personally found the emphasis on connecting vulnerability and dimensionality over-emphasized, in the title, the abstract, and the main narrative. I believe that the most interesting contribution of the paper is understanding the impact of initialization on the vulnerability of the resulting classifier. In my opinion, a title "The impact of initialization on adversarial vulnerability" (and a narrative focusing more on this aspect) would have resulted in a very different discussion here, as well as a different perception of the paper. But again, this is my personal opinion and has nothing to do with whether I think the paper should be accepted or not.

---

> > > > > ### Comment · AnonReviewer3 · 2018-12-03
> > > > > **Further discussion**
> > > > >
> > > > > I second that the dependence on dimension is over-emphasized and the current title is misleading. Based on the experiments in other work, it still seems like the effect of dimension vanishes for adversarially trained models.
> > > > >
> > > > > "In fact I think that plotting gradient norms over training is _crucial_ to support the argument that the situation at initialization is predictive of the situation at convergence."
> > > > >
> > > > > While I find Figure 4 interesting, I'm still not convinced of the argument. To really tie initialization to final performance the authors should show what happens as they change the variance of the weight initialization, if you purposely initialize poorly/better do you get even worse/better sensitivity at test time? Finally, showing the interaction with adversarial training or Gaussian data augmentation is crucial if the authors wish to claim that better initialization strategies can improve upon adversarial training.

---

> > > > > ### Author Response · Authors · 2018-12-10
> > > > > **Thanks for your comment + clarification of our primary goal**
> > > > >
> > > > > We thank your for your comment and accurate analysis, how some formulations may have mislead the interpretation of our paper. We will reformulate any possible over-statements adequately for the final version.
> > > > >
> > > > > We agree that the new experiments suggest that the prior- and post-training vulnerability are related. However, let us stress that proving this relation was not the ambition of this paper. Our primary goal is simply to point out that a/ the prior vulnerability exists, scales like \sqrt(d) and is independent of the net-architecture and the dataset; and b/ that the same dimension dependence can be observed after usual training. Our title does not lie on these observations: our empirical experiments do support that vulnerability increases with usual training, which *is* the standard/common/default training method. But, if it helps, we are willing to change our title to "*Prior* adv. vuln. of n.n. increases with input dim.".
> > > > >
> > > > > More generally, since the paper is already long, and a/&b/ already novel and non-trivial results/observations, we see the present controversy about their implications rather as pro than a cons for publication. The debate attests that our results indeed yield non-trivial new questions for the community to answer.
> > > > >
> > > > > We will be happy to include this clarification in our final version.

---

### Author Response · Authors · 2018-11-19
**Overall answer to reviews**

First, we would like to thank all the reviewers and contributors to the discussions. We are happy to see that our study raises so many questions.  We are in fact a bit surprised to be caught in such a storm, especially since nobody seems to disagree about the facts at a technical level...

We do believe our paper offers the most precise study of the effects of first-order adversarial perturbations; especially, proving independence from network structure, and re-emphasizing the direct relationship with input dimension. The limitations are clearly stated in the text. We do maintain there is a connection with adversarial attacks: if there is a vulnerability at first order, then there is a vulnerability. And we thank AnonReviewer4 for pointing out that many people do care about first-order worst-case perturbations. However, we agree that not all vulnerability comes from first-order phenomena.

There seems to be an ongoing debate about the best terminology for the objects studied in our paper. We are happy to term them first-order inputs perturbations, first-order worst-case corruption, first-order adversarial perturbations, or anything similar that makes sense. We will be happy to receive suggestions and edit the text accordingly.  Would that be acceptable?

As to why our insights on prior vulnerability is relevant to understand the vulnerability of SOTA robust models: as mentioned by AnonReviewer3 citing [2], SOTA models still don't manage to reduce gradients on out-of-training examples (see point 5/ below). This is why we think our study is still highly relevant even after strong adversarial training. (Also see our separate thread on why prior vulnerability matters.)

As for technical remarks:

1/ More defences closer to SOTA, especially, iterative ones: the version updated today covers PGD iterative defences (see Figs. 2 & 6). The conclusions are unchanged.

2/ Concerning Fig 4b of the later work [1]: Fig 4b of [1] is *after* robust adversarial training; our claims refer to before adversarial training. So [1] does not refute our claims: rather, it confirms that robust adversarial training operates via reducing gradient issues.
More precisely, to be comparable across dimensions, the x-axis of Fig 4 of [1] should be rescaled to epsilon / sqrt(d), because their epsilon is measured in l_2-norm rather than l_infty (see our Eq. (3), or their text).  After proper rescaling, Fig 4a shows that naturally trained nets are more vulnerable with growing input-dimension. This confirms our own theorems and experiments (our Fig. 3). After rescaling, the curves of Fig 4b seem to overlap: indeed this suggests that our predicted dimension-dependence vanishes *after* robust adversarial training (rather than usual training). This does not contradict our results, it comforts and completes them: usual training does not escape the prior's vulnerability, and in some situations, ill-behaved priors can be escaped by robust training.

3/ Initialization: most people use He-like initializations for a very good reason: it is essentially the only way to obtain bounded activations at init and be symmetric over the inputs of each unit. Our work thus underlines a conflict between keeping reasonably-valued activations, and having reasonably-sized input gradients. This certainly suggests future studies of other initializations. Such initializations will somehow have to break symmetry between units, and thus introduce implicit priors. Some possibilities would be to favor low frequencies in the image, or to prioritize the learning of some units wrt others by initializing them to larger weights (thus implicitly telling the network to try the large-activation units first, which in effect is a prior on the number of units). We have started to play with this, but this is a whole new research direction, while the present text is long enough...

4/ Concerning "adversarial training eventually gives robustness to the training set, but this robustness does not generalize to the test set [2]. For these models, the gradients are well behaved local to the training points (...) but the gradients aren’t well behaved for new iid samples from the data distribution."
This does not contradict our theory/experiments. On the contrary: it appears that outside the training points where gradients have been decreased, the network might keep the gradient properties from its prior, namely, naturally large gradients. This example hence rather shows that our findings on priors and the resulting gradients should be kept in mind.

Once more, we would like to thank the reviewers for the rich debate. We hope that the technical points have been addressed. We do believe our results are non-trivial and relevant. It seems that the "adversarial" terminology raises issues: we are open to consensus suggestions on this point.

[1] Are adversarial examples inevitable, anonymous, under review for ICLR 2019

---

> ### Author Response · Authors · 2018-11-19
> **Why prior vulnerability matters (compendium to the overall reply)**
>
> The reviewers acknowledge that our results --in particular our theorems and their empirical verification in Fig 3-- are novel and interesting. But some of them wonder how understanding the vulnerability of networks at initialization or after usual training helps understanding their vulnerability after (SOTA-) robust training. We try to answer these concerns with the following two points:
>
> a/ Why understanding prior-vulnerability of neural networks may partly explain their vulnerability after robust training.
>
> Without the right priors, even the best learning algorithms fail.
> For example, if instead of using carefully designed CNNs, we used fully connected (FC) nets for image classification, we'd get much worse accuracies; especially if our FC net had the same amount of neurons than the CNN (although, theoretically, it could learn the same classifier than the CNN). The situation for adversarial vulnerability is the same: of course, smart training algorithms can help to improve robustness. And the fact that the classifiers get more robust shows that they do help up to some extent. But very ill-behaved priors (i.e. naturally vulnerable priors) will certainly harden their task.
>
> The example pointed out by AnonReviewer3, about recent work [2] showing that adversarial training is efficient on the training set but usually fails to generalise to the test set, may actually be evidence in our favour. Think about FC nets again: they typically get 100% accuracy on the training set, but completely fail to generalise to the test set. The fact that this (essentially) does not happen to CNNs --despite using the same training algorithm!-- shows the importance of choosing well-behaved priors on our architectures/classifiers. Of course, one can argue, as in [2], that with more data, this would not happen, and hence, that adversarial vulnerability is essentially a sample-size problem. But an alternative way to get robust classifiers with the same amount of data might be to use better priors, i.e. more carefully designed architectures. Which brings us to our second point.
>
> b/ Our paper essentially shows that our priors are already adversarially vulnerable and usual training does not escape these priors. But how can we use those insights? What "actionable" insight does it give us?
>
> For one thing, as argued in a/, pointing out the vulnerability of our priors gives the community a clear reason to search for better priors, so as to complement our robustification algorithms. But they give even more: our theorems, with their clear assumptions and proofs, may actually guide this search. We may for example ask: how can I design an architecture/block that escapes our assumptions? Or: what essential parts of the proofs actually "generate vulnerability"? Appendix B goes in that direction: noticing that overly large weights are an essential reason for vulnerability, it analyses what happens if we introduce average-pooling layers (that have weights of size 1/d rather than 1/sqrt(d)). This section is still preliminary (which is why it is in appendix), but it illustrates how our results, despite being only on priors, can yield concrete, actionable results to improve adversarial robustness.
>
> [2] Adversarially Robust Generalization Requires More Data, Schmidt et al., 2018

---

> > ### Comment · AnonReviewer3 · 2018-11-21
> > **Regarding the adversarial terminology**
> >
> > The criticism on the study of lp adversarial examples is nuanced and I wanted to clarify what I meant. Security motivations aside, it's been shown that the phenomenon of small worst case perturbations is very closely linked to the phenomenon that vision models are sensitive to large average case perturbations [1]. Indeed the same linearity assumptions made in this work directly link the two phenomena --- for a linear model if I tell you the distance to the decision boundary I have also told you the error rate in the presence of large additive Gaussian noise. Obtaining models which are robust to average case image corruptions is of course desirable (they will actually occur in deployment). Perhaps the reason adversarial researchers don't consider robustness to average case image corruptions is they find it less "intriguing", but this is ultimately a subjective opinion. The authors mentioned that we can't hope to understand robustness to large corruptions if we don't first understand robustness to small corruptions. To flip this around it's also true that we certainly won't be robust in the worst case if we aren't first robust in the average case.
> >
> > All I'm recommending researchers consider measuring is how their theory and defenses are making progress towards generalization outside the natural data distribution, because that ultimately is what this phenomenon is. Enforcing smoothness priors seems like an interesting and potentially useful approach towards this goal, why not demonstrate the usefulness of this prior by showing improved generalization (as measured by test error in corrupted image distributions) rather than *only* measuring the smoothness of our learned functions? ML has been around for a while, and smoothness has long been proposed as an important model prior, but I don't understand why we suddenly see smoothness itself as the goal rather than a means towards better and more robust models.
> >
> > 1. https://arxiv.org/abs/1608.08967

---

> > > ### Author Response · Authors · 2018-11-23
> > > **Answer regarding adversarial terminology**
> > >
> > > Thank you for your quick reactions.
> > >
> > > We understand that you have different intuitions and preferences than us about this problem. But your own reference [1] suggests that both views (vulnerability to small worst-case perturbations as opposed to larger average perturbations) are essentially equivalent. We will be happy to mention that.
> > >
> > > More generally, do you believe that you should oppose the publication of our paper because you would have written a different one?  In our opinion, a diversity of viewpoints is a good thing.

---

> > > > ### Comment · AnonReviewer3 · 2018-11-27
> > > > **Thanks for the response**
> > > >
> > > > I appreciate the authors engaging in this important discussion. I think it's important for the introduction of papers to properly discuss the motivation of their work. Your introduction alludes to adversaries constructing imperceptible perturbations to fool statistical classifiers, as if this is a realistic concern to be dealt with. I see limited practical impact for "defending" against imperceptible perturbations. If an adversary wants to fool a statistical classifier, in most settings they can do far worse than tiny changes to the input.
> > > >
> > > > If your motivation is instead methods for learning smoother functions because you believe it will improve model generalization or robustness to distributional shift, write that and evaluate accordingly.
> > > >
> > > > As I wrote in my review, I'm willing to evaluate the technical details of your work and indeed I see model smoothness as a useful prior for improving more general notions of robustness. While I think your work would be strengthened had you directly considered more realistic forms of robustness, if you are suggesting that my score is purely because your definition of model robustness has limited practical motivation, that is not true.

---

### Author Response · Authors · 2018-11-28
**Summary of paper updates**

We thank the reviewers for their stimulating comments that motivated the changes below, which strengthen the paper. We hope that this new version fully addresses the reviewers' concerns by highlighting more clearly the value and limitations of our results on prior vulnerability.

(1st update)

On Fig. 2, we added PGD training with random starts. The new curves corroborate the validity of our first-order analysis. Overall conclusions remain unchanged.

(Latest update)

We added a paragraph (Sec 5.1) discussing the implications of our results on prior vulnerability. It relies on a new figure (Fig. 4) that reveals an unmistakable discrepancy between the gradient-norm evolution on the training and test sets respectively. This suggests that nets tend to recover their prior gradient properties outside the training sample, which in turn calls for new architectures with smaller prior gradients and strengthens the importance of our analysis of current priors.

This new figure required to re-do the experiments on the dimension-dependence of vulnerability and gradients (Sec. 4.2), which therefore got updated as well (Fig. 3). To accelerate training, we used upsampled CIFAR-10 datasets, and pushed the previous results on downsampled ImageNet subsets to Appendix E. Conclusions are unchanged.

The related literature section (§ On network vulnerability) now mentions reference [1] (pointed out by AnonReviewer3), which links small worst-case perturbations to larger average perturbations.

Finally, we rephrased our conclusion to reflect more accurately the implications and limitations of our results.

[1] Robustness of classifiers: From adversarial to random noise, Fawzi et al., NIPS, 2016

---

### Author Response · Authors · 2018-12-10
**Overall summary and clarification of our paper's current primary goal**

We thank all reviewers for their comments and implication in the review process and seize this opportunity for a final recap on our paper and its contributions.

First, our work re-emphasises the strong link between small adversarial perturbations and gradient norms. Although this link was known, many still believe that simple gradients cannot explain f. ex. the vulnerability to iterative methods. Our experiment on Fig.2 with PGD trained networks clearly suggests otherwise.

Second, and more importantly, we are first to point out the prior vulnerability of classical networks, its exact scaling with the input dimension, and its relative independence of the architecture and dataset. We are also first to precisely measure and confirm this dimension-dependence on trained networks, using usual training and real-world, classical datasets.

Our results however do not precisely study *how* prior-vulnerability acts on post usual (robust) training vulnerability. Nor do they prove that no robust training method will ever be able to robustify our current networks. It is not their current ambition. We will clarify this in the final version and rephrase possible over-statements adequately. However, the current results do *suggest* that architectures with less vulnerable priors might be easier to robustify; so much so, that we are now mainly being reproached not to have dug further in these directions.

We agree that these are thrilling questions. But since the paper is already long, and its afore-mentioned contributions already novel and non-trivial, we see the present controversy about its implications rather as a pro than a cons for publication. The debate attests that our results do indeed yield non-trivial new questions to be answered.

We therefore call upon all reviewers and the AC, for the final decision, to focus less on what hasn't been done, and more on our actual contributions and the new perspectives that they open for future research.

---

### Public Comment · ~Adam_M_Oberman1 · 2019-03-10
**About dimensional scaling**

When comparing images at different resolution, it is important to scale the units of measurement.  This is common practice in mathematics and physical science, but is often not done in computer science.

For example, consider a square of side length 1/2 centered in a 1 by 1 box.  The area of the square is clearly 1/4.  At resolution 32x32, it makes sense to multiply lengths by h = 1/32 and volumes by h^2.   Then if we consider resolution 128X128, we keep the same units.

Dimensional scaling will affect the conclusions of this article about the size of perturbations.  For example, consider a small physical perturbation on a patch of size \delta x \delta with average size \epsilon.    In scaled units, the norm of the vector is nearly same, at all resolutions.   On the other hand, in unscaled units, it appears to be a larger vector at higher resolution.  But the size of the original image vector is also larger, by the same proportion.

Specifically, the image had 4 times as many 1s at higher resolution, and the perturbation has 4 times as many components of size \epsilon.
So the relative size of the perturbation didn’t change:

Main point: since the norm of the images as well as the norm of the perturbations grows with the dimensions, when we measure the size of the perturbation in non-dimensional units, it does not change.

See https://en.wikipedia.org/wiki/Dimensional_analysis for more information.

---

> ### Author Response · Authors · 2019-03-10
> **Dimensional scaling is accounted for**
>
> Thank you for your comment. Perturbation scaling has been taking into account in our analysis. See paragraph "Calibrating the threshold \epsilon to the attack-norm." in Sec.2 and see Appendix D.
>
> The idea is essentially equivalent to what you explain: keep the signal to noise ratio (i.e. perturbation norm over image norm) constant across dimensions. In l-infinity norm, this leads to a constant perturbation threshold \epsilon_\infty; in l2-norm, this leads to a threshold epsilon_2 proportional to \sqrt d; and in lp-norm proportional to d^{1/p}. That is why the x-axis of Figs 1a-c are scaled by d^{1/p}.
>
> Note that our main theorem shows that adversarial damage increases like sqrt d, but only after adjusting the perturbation threshold.
> More precisely, it shows that |dL(x)|_q scales like d^{1/q - 1/2}. So, with perturbation threshold epsilon_p of lp-attacks scaling like d^{1/p}, we get
>
> 	adv. dam. := max_{|\delta|_p \leq \epsilon_p} L(x+\delta) - L(x)
> 		  \approx \epsilon_p |dL(x)|_q
> 		  = \sqrt d,
>
> where 1/p + 1/q = 1.
>
> (Note that using p=infty and q=1 has the "advantage" to yield a gradient norm |dL|_1 that is proportional to adversarial damage.)

---

### Meta-Review · Area_Chair1 · 2018-12-15
**The paper has intriguing ideas but requires more work**

**Confidence:** 5
**Recommendation:** Reject

**Metareview:**

This paper suggests that adversarial vulnerability scales with the dimension of the input of neural networks, and support this hypothesis theoretically and experimentally.

The work is well-written, and all of the reviewers appreciated the easy-to-read and clear nature of the theoretical results, including the assumptions and limitations. (The AC did not consider the criticisms raised by Reviewer 3 justified. The norm-bound perturbations considered here are a sufficiently interesting unsolved problem in the community and a clear prerequisite to solving the broader network robustness problem.)

However, many of the reviewers also agreed that the theoretical assumptions - and, in particular, the random initialization of the weights - greatly oversimplify the problem. Reviewers point out that the lack of data dependence and only considering the norm of the gradient considerably limit the significance of the corresponding theoretical results, and also does not properly address the issue of gradient masking.